# The m⁶A pathway protects the transcriptome integrity by restricting RNA chimera formation in plants

Dominique Pontier[1,2,*], Claire Picart[1,2,*], Moaine El Baidouri[1,2], François Roudier[3], Tao Xu[4], Sylvie Lahmy[1,2], Christel Llauro[1,2], Jacinthe Azevedo[1,2], Michèle Laudié[1,2], Aurore Attina[5], Christophe Hirtz[5], Marie-Christine Carpentier[1,2], Lisha Shen[6], Thierry Lagrange[1,2]

Global, segmental, and gene duplication–related processes are driving genome size and complexity in plants. Despite their evolutionary potentials, those processes can also have adverse effects on genome regulation, thus implying the existence of specialized corrective mechanisms. Here, we report that an N6-methyladenosine (m⁶A)–assisted polyadenylation (m-ASP) pathway ensures transcriptome integrity in *Arabidopsis thaliana*. Efficient m-ASP pathway activity requires the m⁶A methyltransferase-associated factor FIP37 and CPSF30L, an m⁶A reader corresponding to an YT512-B Homology Domain-containing protein (YTHDC)-type domain containing isoform of the 30-kD subunit of cleavage and polyadenylation specificity factor. Targets of the m-ASP pathway are enriched in recently rearranged gene pairs, displayed an atypical chromatin signature, and showed transcriptional readthrough and mRNA chimera formation in FIP37- and CPSF30L-deficient plants. Furthermore, we showed that the m-ASP pathway can also restrict the formation of chimeric gene/transposable-element transcript, suggesting a possible implication of this pathway in the control of transposable elements at specific locus. Taken together, our results point to selective recognition of 3′-UTR m⁶A as a safeguard mechanism ensuring transcriptome integrity at rearranged genomic loci in plants.

## Introduction

N⁶-methyl-adenosine (m⁶A) has recently emerged as a prevalent mRNA modification that tends to be enriched in 3′-UTR within the consensus RRm⁶ACH motif (where R is A/G and H is A/C/U) (Dominissini et al, 2012; Meyer et al, 2012; Schwartz et al, 2013; Shen et al, 2016). m⁶A mRNA modification occurs co-transcriptionally and is driven by a conserved writer complex composed of methyltransferase-like3 (METTL3/MT-A70), METTL14 (a noncatalytic METTL3-homologous protein), and Wilms tumor 1-associated protein (WTAP/FIP37) (Bokar et al, 1997; Zhong et al, 2008; Liu et al, 2014a; Shen et al, 2016; Meyer & Jaffrey, 2017). Mutation in any of these genes prevents m⁶A methylation and causes detrimental effects leading to differentiation defects and apoptosis in animal cells (Bastita et al, 2014; Geula et al, 2015), embryonic lethality in plants, and mice (Zhong et al, 2008; Geula et al, 2015; Shen et al, 2016; Růžička et al, 2017), defects in meiosis in yeast (Schwartz et al, 2013), and defects in gametogenesis and sex determination in insects (Hongay & Orr-Weaver, 2011; Haussmann et al, 2016).

m⁶A is expected to control the fate of mRNA mostly by recruiting specific readers that belong to the YT521-B homology (YTH) domain-containing proteins and bind m⁶A methyl group through a conserved hydrophobic pocket (Meyer & Jaffrey, 2017; Wu et al, 2017). YTH proteins fall into three major groups defined as the YTHDF- and YTHDC2-type clades whose members are primarily cytoplasmic and the YTHDC1-type clade whose members are expected to act in the nucleus (Meyer & Jaffrey, 2017). Recent studies performed in animal and plant models have connected m⁶A-dependent recruitment of YTH-type proteins to the control of various aspects of mRNA metabolism, including stability (Wang et al, 2014; Wojtas et al, 2017), splicing (Lence et al, 2016; Xiao et al, 2016), alternative polyadenylation (APA) (Ke et al, 2015; Yue et al, 2018), nuclear export (Roundtree et al, 2017), and translation initiation (Wang et al, 2015). More recently, m⁶A modification has been also shown to play a role in XIST-mediated X-chromosome inactivation (Patil et al, 2016), repair of UV-induced DNA damages in mammals (Xiang et al, 2017), and response to abiotic stress in plants (Anderson et al, 2018), confirming its broad spectrum of activities in eukaryotes.

Compared with other eukaryotic genomes, plant genomes show high dynamics with a substantial degree of diversity (Rensing, 2014). Genome and gene/transposon duplication-related processes in plants are now regarded as important evolutionary forces that contribute to genome complexity, generating new genetic material as

[1]Centre National de la Recherche Scientifique, Laboratoire Génome et Développement des Plantes, UMR 5096, Perpignan, France [2]Univ. Perpignan Via Domitia, Laboratoire Génome et Développement des Plantes, Unité Mixte de Recherche 5096, Perpignan, France [3]Laboratoire Reproduction et Développement des Plantes, Univ Lyon, Ecole Normale Supérieure de Lyon, Université Claude Bernard Lyon1, Centre National de la Recherche Scientifique, Institut National de la Recherche Agronomique, Lyon, France [4]Department of Biological Sciences, National University of Singapore, Singapore [5]Platform SMART/Laboratoire de Biochimie et Protéomique Clinique/ Plateforme de Protéomique Clinique, University of Montpellier, Institut de Médecine Régénérative et de Biothérapie , Centre Hospitalier Universitaire Montpellier, Institut national de la santé et de la Recherche Médicale, Montpellier, France [6]Temasek Life Sciences Laboratory, 1 Research Link, NUS, Singapore

Correspondence: lagrange@univ-perp.fr
*Dominique Pontier and Claire Picart contributed equally to this work as first authors

well as nonfunctional sequences that are rapidly eliminated (Rensing, 2014; Panchy et al, 2016; Galindo-González et al, 2017). Studies of naturally occurring genomic rearrangements in plants revealed that duplicated/translocated loci can impact initial gene regulatory networks in *cis* or *trans*, resulting either in beneficial, deleterious, or neutral effects depending on the genetic context. Whereas *trans*-acting effects on original genes generally entail gene silencing through natural epigenetic variation (Bender & Fink, 1995; Durand et al, 2012), *cis*-acting effects on a neighboring gene vary among cases, leading to epigenetic control (Zheng & Cheng, 2014; El Baidouri et al, 2018), chimeric gene formation (Thimmapuram et al, 2005; Shahmuradov et al, 2010), polyadenylation defects (Tsukamoto et al, 2010), transcriptional interference (Kashkush et al, 2003), and gene/exon transduction (Xiao et al, 2008; Zhu et al, 2016).

There is compelling evidence that plants develop specific control mechanisms to neutralize the adverse effects that can result from loci rearrangements. For instance, genetic studies revealed that the cell cycle regulator gene *BONSAI* (*BNS*) is insulated from the DNA- and histone-dependent repressive effects of an adjacent LINE retroelement by the action of *INCREASE IN BONSAI METHYLATION1*/*IBM1*, a jmjC domain-containing histone demethylase (Saze et al, 2008). Likewise, a pathway comprising plant-specific RNA- and chromatin-binding proteins has been implicated in the control of transcriptome integrity by counteracting premature transcription termination defects caused by the presence of an intronic transposable element (TE) (Wang et al, 2013a; Saze et al, 2013). Given the importance of gene duplication/translocation mechanisms for gene expansion in plants, it is likely that plants develop additional mechanisms to neutralize potential detrimental effects of such gene rearrangements. Here, we show that m6A RNA modification and a network of conserved and plant-specific proteins constitute a post-transcriptional regulatory pathway that enforces polyadenylation and 3′-end formation at loci that exhibit transcriptional readthrough, cryptic intergenic splicing, and chimeric mRNA formation in *Arabidopsis thaliana*. Targets of this newly identified pathway are enriched in rearranged gene pairs (GENE1/GENE2) that present duplicated or translocated paralog/pseudogene at GENE2 position, suggesting that gene rearrangements at targeted loci could be responsible for the observed termination/polyadenylation defects. We show that GENE1 transcripts are m6A-tagged and that m6A-assisted polyadenylation at these genes requires the concerted action of the m6A methyltransferase auxiliary factor FIP37, and CPSF30L, a plant-specific YTH-domain containing variant of the conserved 30-kD subunit of cleavage and polyadenylation specificity factor (CPSF30). Finally, we provide evidence that the m-ASP pathway can also restrict the formation of potentially deleterious chimeric GENE-TE transcripts. Taken together, this study establishes a strong link between m6A methylation and the control of transcriptome integrity in plants, expanding the repertoire of functions performed by this type of RNA modification in eukaryotes.

## Results

### NERD mutations lead to chimeric mRNA formation in *A. thaliana*

We previously reported that *NERD* gene mutations relieve DNA methylation–dependent repression of a subset of newly acquired loci in *A. thaliana*, including a pseudogene named as *pseudo-ORF* (*psORF*) (Pontier et al, 2012). Investigating further NERD's function, we found evidence of cotranscriptional readthrough andchimeric RNA formation between *psORF* and a Copia-type transposon (AT5TE50260) located downstream (Fig S1A). The accumulation of chimeric RNA species in *nerd-1* was validated by semi-quantitative RT–PCR assay using primers anchored on both sides of the *psORF*/AT5TE50260 genomic junction as well as by real-time qRT–PCR assay with primers specific for *psORF* and AT5TE50260 loci (Fig S1B and C). Interestingly, although the expression of a flag-tagged version of NERD (*nerd*+T) did not restore *psORF* silencing in *nerd-1*, as previously reported in Pontier et al (2012), it efficiently reverted the read-through phenotype (Fig S1A–C), suggesting a contribution of NERD in transcription termination control at *psORF* locus.

To investigate whether NERD controls transcription termination and chimeric RNA formation at loci other than *psORF*, we scrutinized our *nerd-1* transcriptome data and observed several cases of mRNA chimera formation. In many instances, as exemplified by AT4G30570 and AT1G71330 (GENE2), increased mRNA levels spread across the upstream intergenic region as a possible result of defective RNA 3′-end formation at upstream AT4G30580 and AT1G71340 genes (GENE1), respectively (Fig S2A). The specific accumulation of chimeric mRNA species in *nerd-1* was validated by RT–PCR assay using specific GENE1/2 (F1/R1 and F2/R1) primers and RNA gel blot analysis using GENE1-specific probes (N) (Fig S2B and C). The readthrough phenotype was specific to NERD deficiency as it was reverted in *nerd*+T plants (Fig S2A–C). Consistent with a role for NERD in transcription termination control at GENE1 rather than transcription control at GENE2, we did not detect any enrichment of H3K4me3, a chromatin mark associated with active promoter regions, in the corresponding intergenic regions in *nerd-1* compared with WT or *nerd*+T plants (Fig S2D).

To evaluate the generality of the observed readthrough pattern, we calculated the variation in strand-specific read density in the 500-bp windows downstream of all annotated genes upon NERD depletion. As an apparent increase in downstream read density might derive from upstream gene deregulation, all genes whose expression showed a significant change in *nerd-1* mutant were also excluded from the list of gene candidates. Using a twofold change cutoff, 107 genes showing unaffected levels of expression in *nerd-1*, but a significant increase in strand-specific read density in their downstream region were identified (Fig 1A and Table S1). Inspection of the candidate loci using the integrative genomics viewer confirmed the existence in *nerd-1* mutant of numerous transcription readthrough events leading in most cases to the formation of putative mRNA chimera (GENE1/GENE2) but also of mRNAs with extended 3′-UTR (GENE1e) (Table S2). To ascertain that annotated GENE2s correspond to bonafide genomic units rather than misannotated 3′ region of GENE1s, we assessed the conserved collinearity of *A. thaliana* GENE1/GENE2 chimeric loci (excluding tandem repeats that have a clear evolutionary origin) in 325 sequenced angiosperms species, including monocots and dicots (Fig 1B). This comparative genomic analysis showed that a colinear GENE1–GENE2 organization was not conserved in most cases, with many species having only one member of the gene pair present in their genome (G1 only or G2 only), or both genes present but at distant genomic locations (Fig 1B), therefore supporting the fact

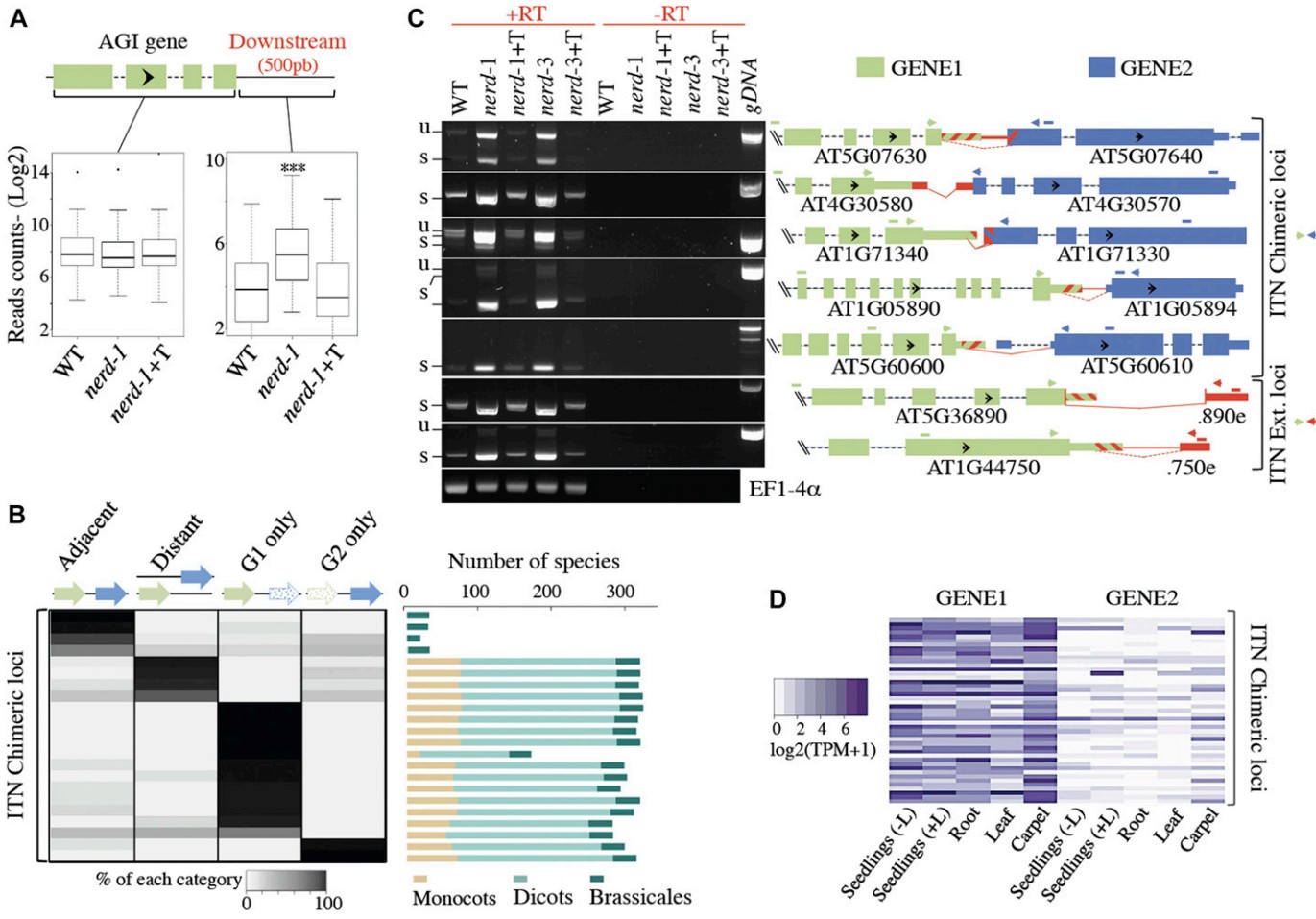

**Figure 1. NERD restricts chimeric mRNA formation in *Arabidopsis*.**
**(A)** Boxplot showing the read counts in gene body and corresponding downstream region for genes showing enhanced transcription readthrough in *nerd-1*. ***$P < 0.001$; *t* test. **(B)** Evolutionary analysis of GENE1/GENE2 conservation in 325 angiosperm genomes. The heat map shows the relative percentage of four different scenarios for each gene pair: adjacent: GENE1 and GENE2 have a contiguous conserved position; distant: GENE1 and GENE2 are separated on the same or different chromosomes. GENE1 only and GENE2 only: correspond to situations where only one gene of the tandem pairs was found in the genome. **(C)** Semi-quantitative RT–PCR analysis of extended/chimeric ITN mRNAs in WT, *nerd-1*, *nerd-1*+T, *nerd-3*, and *nerd-3*+T plants (left panel). Exons are shown with colored thick bars, UTRs with colored thin bars, introns with black dashed lines, and *nerd*-dependent cryptic intergenic introns with dashed diagonal red lines. The annotated exons and UTR regions partially spliced in *nerd-1* are indicated by cross-hatched bars. The unannotated exons or UTR regions newly appearing in *nerd-1* are shown with red bars. Primers used in the RT–PCR experiments are shown as arrowheads on the right panel. Transcript regions analyzed in quantitative RT–PCR are labeled as bars. Unspliced (u) or spliced (s) forms of RT–PCR products corresponding to chimeric or extended mRNA are indicated. *EF1-4α* was used as loading control. Minus RT (–RT) reactions are controls for DNA contamination. (right panel) **(D)** Expression score of GENE1 and GENE2 at ITN loci in different tissues using already published RNA-seq data (Araport11; Krishnakumar et al, 2015) and expressed in TPM.

that GENE1 and GENE2 represent independent gene units. Interestingly, the few cases in which a contiguous GENE1–GENE2 organization was phylogenetically conserved corresponded to sequences found exclusively in close relatives of *A. thaliana* (Brassicales) (Fig 1B), indicating a recent evolutionary origin.

To characterize further the chimeric/extended readthrough phenotype, several candidate loci in addition to AT4G30580/70 and AT1G71340/30 were selected for analysis. Using primers anchored on both sides of the corresponding chimeric/extended regions (and thus spanning the intergenic region) in RT–PCR assays, we confirmed the specific over-accumulation of readthrough mRNAs in two independent *nerd* mutant backgrounds (Fig 1C). Sashimi plots indicating RNA-seq reads that specifically cross the GENE1–GENE2 junction in *nerd-1* background are further shown in Figs S2A and

S3A, and B. The specific up-regulation of chimeric/extended mRNA forms in nerd-1 was further validated by real-time quantitative reverse transcription-polymerase chain reaction (qRT–PCR) assay (Fig S3C), confirming that the readthrough events are not a consequence of unleashed GENE1 transcription. Heat map analysis further confirmed that GENE2 at NERD-dependent chimeric loci are generally silent or expressed at a low level in WT plant organs (Fig 1D), likely owing to the termination-promoting activity of NERD. From now on, we will refer to genes showing termination defects in *nerd* mutant backgrounds as ITN (inefficient termination in NERD) loci.

Prior studies have shown that the split ends protein (SPEN) family proteins FPA/FCA repress the formation of chimeric mRNAs at a subset of *A. thaliana* genomic loci (Sonmez et al, 2011; Duc et al, 2013). However, a limited overlap is currently observed between loci

reported to form mRNA chimera in *fpa-7* and those observed in *nerd-1* mutant plants (Fig S4A) (Duc et al, 2013), an observation that prompted us to evaluate the extent to which these proteins interact in vivo. Using intergenic primers in RT–PCR assays, we confirmed that NERD was not required to control extended/chimeric mRNA formation at most FPA-dependent loci tested (hereafter named as FPA-only) (Fig S4B). Reciprocally, FPA activity was not mandatory for the control of chimeric mRNA formation at most ITN loci (Fig S4B). While the mechanisms underlying target specificity of NERD and FPA in this process remain unclear, expression analysis indicates that FPA controls mRNA chimera formation preferentially between two expressed loci (Fig S4C). Taken together, our results reveal the specific and largely independent roles of NERD and FPA proteins in the control of transcription termination and chimeric mRNA formation, further highlighting the existence of multiple chimeric mRNA control pathways in *A. thaliana*.

### Rearranged gene pairs are significantly overrepresented among mRNA chimera-producing ITN loci

To uncover common evolutionary trends among mRNA chimera-producing ITN loci, we traced their evolutionary history in *A.* *thaliana* and its close relatives *A. lyrata* and *Capsella rubella*, after their classification into five categories: single copy genes, segmental/WGD genes, tandemly duplicated genes, translocated duplicates, and finally other duplicates (Fig 2A and Table S3). Translocated duplicates refer to nonsyntenic paralogs that have been relocated elsewhere in the genome since the divergence between *A. thaliana* and *A. lyrata* and/or *C. rubella*. They may originate from single gene duplication or from rearrangements that occurred after the paralog duplication. As shown in Fig 2A both GENE1 and GENE2 are represented in all categories indicating that, overall, they have various evolutionary origins, being either single or duplicated genes. However, it is interesting to note that both GENE1 and GENE2 are enriched in tandem duplicates (GENE1, *P*-value < 0.0001; GENE2, *P*-value < 0.001) in comparison with the total gene set in *A.* *thaliana* (Fig 2A). In addition, translocated paralogs are overrepresented among GENE2 compared with both GENE1 and total gene sets (*P*-value < 0.001) (Fig 2A and Table S3). Figs 2B and S5 give two examples of translocated GENE2. The first one, AT5G60610, corresponds to a translocation that is present in *A. thaliana* only, whereas AT1G71330 comes from a translocation that occurred in the last common ancestor of *A. thaliana* and *A. lyrata*, after the separation from the *C. rubella* lineage, and which was followed by a

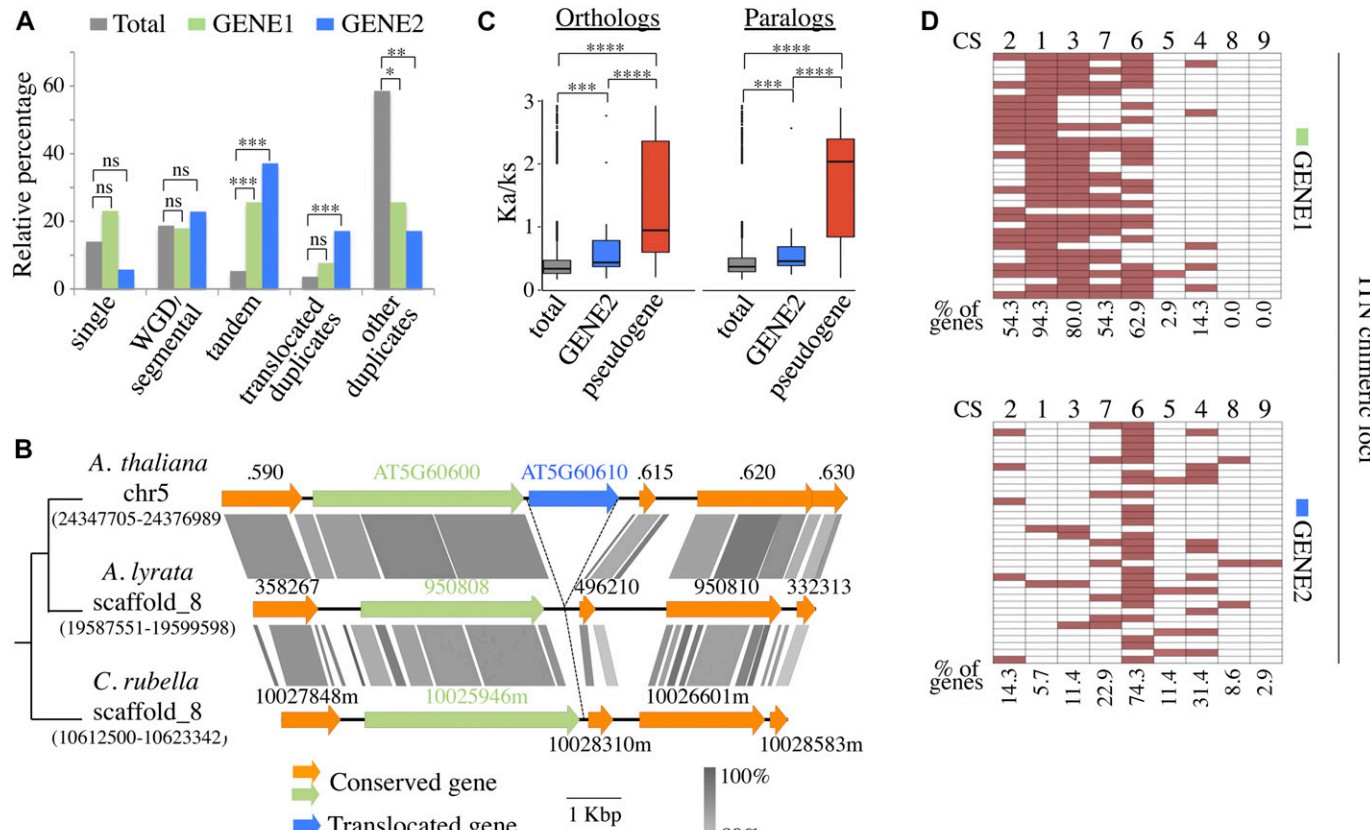

**Figure 2.  Recently rearranged gene pairs are significantly overrepresented among ITN loci.**
**(A)** Relative percentage of GENE1/GENE2 duplication categories compared with total gene set in *A. thaliana*. **(B)** Examples of GENE2 translocation that occurred recently in *A. thaliana* (top), after the separation from the *A. lyrata* (middle), and *C. rubella* (bottom) lineages. **(C)** Boxplot of Ka/Ks ratios of GENE2, pseudogenes, and total genes using closest paralogs in *A. thaliana* and orthologs in *A. lyrata*. In (A) and (C), *P*-values were determined using Kruskal–Wallis statistical test between each gene category. ns: not significant; *P* < 0.05, **P* < 0.01, ***P* < 0.001, and ****P* < 0.0001. **(D)** Distribution of GENE1 and GENE2 from ITN loci among the nine CSs defined in *A.* *thaliana*. The percentage of genes associated with each CS is indicated.

more recent rearrangement. Altogether, these data suggest that recently rearranged gene pairs are significantly overrepresented among mRNA chimera-producing ITN loci.

Low expression of GENE2 at ITN chimeric loci may be a hallmark of pseudogenization, which is a common outcome of gene duplication/translocation in plants (Rensing, 2014; Panchy et al, 2016; Galindo-González et al, 2017). Thus, we estimated the Ka/Ks ratio of GENE2 in comparison with both total gene set and previously identified pseudogenes (Krishnakumar et al, 2015). A Ka/Ks ratio approaching 1 is indicative of the absence of selective constraint. As shown in Fig 2C, GENE2s have significantly lower Ka/Ks values compared with well-characterized pseudogenes. Despite this clear difference with pseudogenes, GENE2 loci present higher Ka/Ks in comparison with the *A. thaliana* total gene set that may indicate ongoing pseudogenization.

In addition, we surveyed the chromatin signatures of GENE1/ GENE2 at ITN chimeric loci in WT seedlings using the nine prevalent chromatin states (CSs) previously defined in *A. thaliana* (Sequeira-Mendes et al, 2014; Vergara & Gutierrez, 2017). In agreement with their expression level, GENE1s were associated with CSs characterizing active transcription units, including CS2, 1, 3, 7, and 6 that correspond to proximal promoters, transcription start site, 5′ end of genes, long coding sequences, and 3′ end of genes, respectively (Fig 2D). Remarkably, the corresponding GENE2s were significantly depleted in the CSs bearing the hallmarks of transcription initiation such as H3K4me3 or the presence of the H3.3 variant and were preferentially enriched in CS6 (Fig 2D and Table S4). This CS is found at the 3′ end of genes and characterized by a slight enrichment in the H2AZ variant (Sequeira-Mendes et al, 2014) and by the presence of H3K36me3, a chromatin modification associated with transcriptional elongation (Vergara & Gutierrez, 2017). Such a dichotomous distribution of chromatin signatures at GENE2 was not observed at a genome-wide level when considering all gene pairs harboring a GENE1 in CS1 (13,988 cases: Chi-square = 86.414, $P$-value = 2.4 × 10$^{-5}$), whereas GENE2 at FPA-dependent chimeric loci followed the genome-wide trend (Chi-square = 7.498, $P$-value = 0.440) (Fig S4D and Table S4).

The atypical chromatin and Ka/Ks signatures of GENE2s raise questions about their potential protein-coding capacity. A mass spectrometry (MS)–based proteomics study of proteins extracted from either WT or *nerd*-1 plants failed to reveal any peptides corresponding to putative GENE2-encoded proteins, possibly because of low expression levels or lack to obtain in-depth protein identification from total extracts (data not shown). To circumvent this limitation, we developed a sensitive reporter system to visualize GENE2 translation in vivo. We generated two transcription readthrough ITN reporters (AT4G30580/70m and AT1G71340/30m) where a Myc-tag sequence was fused in frame at the C-terminal of the GENE2 ORF (Fig S5B). Although showing similar readthrough phenotype in *nerd*-1 (Fig S2A), both reporters are expected to differ in their GENE2 protein-coding capacity as AT1G71330, in contrast to AT4G30570, loses the ATG initiator codon upon chimeric mRNA formation (Fig S2A). Upon plant transformations and Western blot analysis, only AT4G30580/70m-tag–transformed plants showed a Myc-tagged signal at the size expected for the AT4G30570m predicted protein (Fig S5B), suggesting that GENE2 may have retained protein-coding capacity. However, primary sequence prediction

indicates that AT4G30570m probably encodes a nonfunctional homolog of the GDP-mannose pyrophosphorylase AT2G39770/VTC1 lacking the first 20 amino acids of a conserved and essential N-terminal nucleotidyl transferase domain (Qin et al, 2008; Li et al, 2016) (Fig S5C). This truncation results in a missed start codon that shifts the predicted initiation codon to the next in frame methionine at position 22 (Fig S5C). Altogether, our data suggest that ITN loci are often rearranged and present a predictive chromatin signature in *A. thaliana*.

## The m$^6$A pathway mutant fip37 exhibits mRNA chimera formation at ITN loci

NERD (needed for RDR2-independent DNA methylation) was previously characterized via a computational screen designed to identify RNA silencing–related proteins owing to the presence of glycine-tryptophane//tryptophane-glycine (GW/WG)-rich ARGONAUTE-anchor (Argonaute [AGO]-hook) motifs (Azevedo et al, 2011; Pontier et al, 2012). To assess the potential role of AGO hook motifs in the termination-promoting activity of NERD, we transformed *nerd-1* mutant with constructs expressing either a WT (*nerd*+T) or an AG/ GA-mutated version of NERD (*nerd*+Tag), previously shown to be defective in binding AGO (Fig S6A) (Pontier et al, 2012). Both T and Tag variants rescued termination defects at all loci tested, suggesting that the AGO hook motifs are not required for the termination-promoting activity of NERD (Fig S6B and C).

To get further insights into the mechanisms involved in mRNA chimera control at ITN loci, we took advantage of the observation that GENE2 at ITN chimeric loci are poorly expressed in WT plants (Fig 1D) and surveyed published transcriptome data for mutation(s) that would lead to global GENE2 up-regulation. We found numerous GENE2s up-regulated in the transcriptomes of two mutant plants deficient in *N*6-adenosine (m$^6$A) mRNA methylation by virtue of a post-embryonic knockout of two conserved players of m$^6$A methyltransferase complex, FKBP12-interacting protein 37 KD (*fip37-4 LEC1:FIP37*: hereafter referred to as *fip37*L; Shen et al, 2016) and MTA (*mtaABI3:MTA*; Anderson et al, 2018) (Table S2), suggesting the possible role of the m$^6$A pathway in the control of mRNA chimera formation at ITN loci. Visual inspection of RNA-seq alignments, sashimi plots, and intergenic RT–PCR assays further support this conclusion showing readthrough transcription and mRNA chimera formation at ITNs, but not at FPA-only loci in *fip37*L mutant (Figs 3A and B and S7). Calculating the changes in read density in the 500-bp windows downstream of annotated genes upon FIP37 depletion revealed a set of 174 genes experiencing readthrough transcription in *fip37*L mutant (Table S5), with a significant ($P$-value < 0.0001; hypergeometric test) overlap with ITN loci (about 59% of ITN loci and 36% of FIP37L-dependent loci) (Fig 3C). Examination of ITN regulatory circuitry using epistasis analysis confirmed a marked increase in extended/chimeric transcripts in *fip37*L compared with *nerd-1* backgrounds and a nonadditive effect of *nerd-1* and *fip37*L mutations (Fig 3D), indicating that FIP37 is epistatic to NERD in the control of mRNA chimera formation at ITN loci.

Consistent with the finding that WTAP/FIP37 is a conserved component of m$^6$A methyltransferase complex in eukaryotes (Meyer & Jaffrey, 2017), FIP37 depletion triggers a transcriptome-wide loss of m$^6$A mRNA modification in *A. thaliana* (Shen et al, 2016). To get further

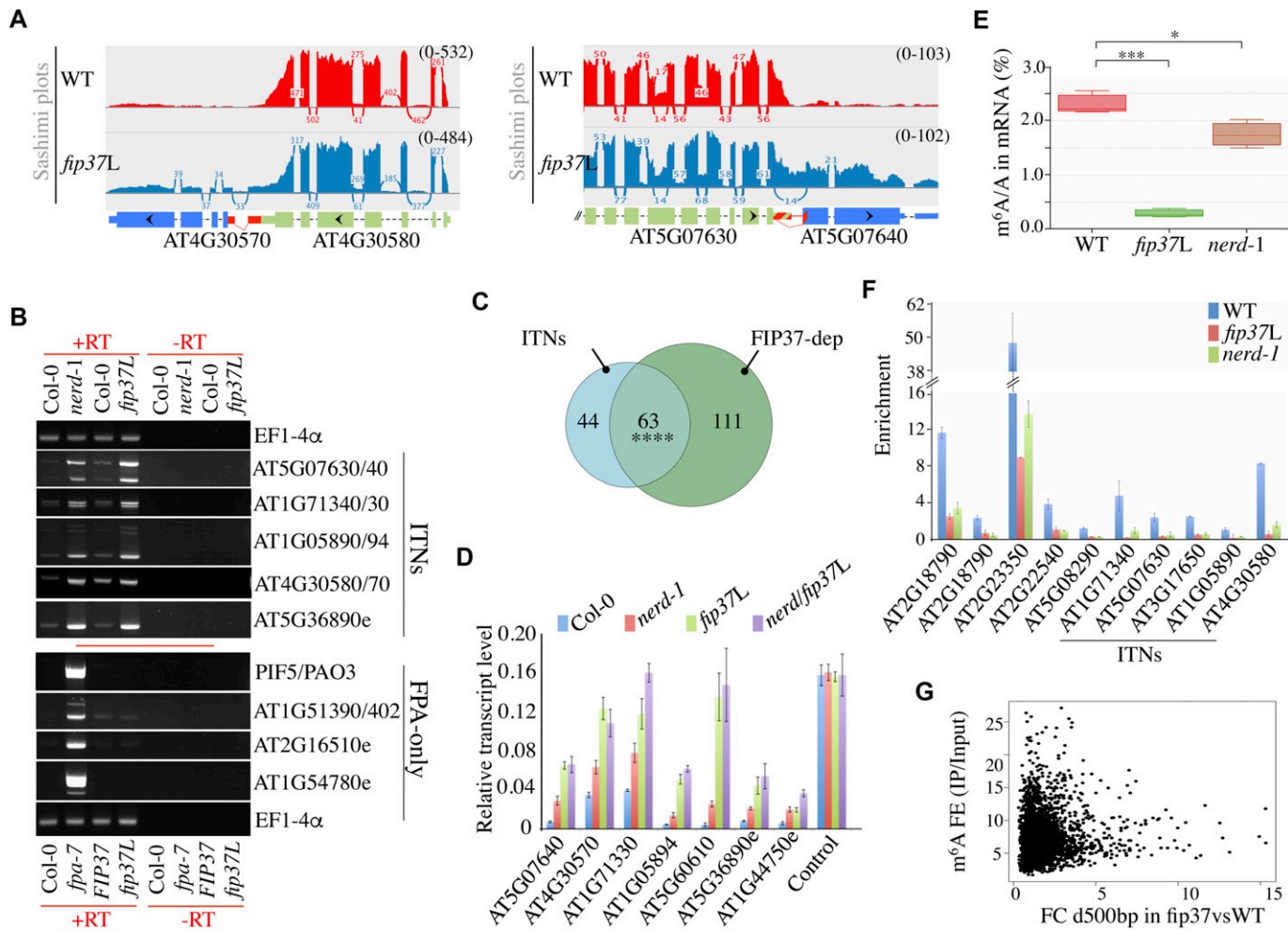

**Figure 3. FIP37-dependent m⁶A modification controls mRNA chimera formation at ITN loci.**
**(A)** Shown are sashimi plots of two ITN loci forming chimeric mRNA transcripts in *fip37*L mutant. The transcript structures and annotations are the same as in Fig 1C. **(B)** Semi-quantitative RT–PCR analyses at ITN loci in *nerd-1*, *fip37*L, and their corresponding WT controls (top part) and at FPA-only loci in *fpa-7*, *fip37*L, and their corresponding controls (bottom part). *EF1-4α* was used as loading control. Minus RT (−RT) reactions are controls for genomic DNA contamination. **(C)** Overlap between loci showing an increase in downstream read counts in *nerd-1 and fip37*L plants. **** denotes *P*-value < 0.0001 for enrichment in the overlap, hypergeometric statistical test. **(D)** qRT–PCR analysis of extended/chimeric ITN mRNAs in WT, *nerd-1*, *fip37*L, and *nerd-1*/*fip37*L plants. WT and *nerd-1* lines used here correspond to segregants harboring the *LEC-FIP37* transgene. Relative transcript levels were normalized to actin using the ΔΔCt method. Data represent the means of four replicate experiments and error bars correspond to SD values. AT4G26410 is used as a control. **(E)** LC-MS/MS quantification of the m⁶A/A ratio in mRNA isolated from WT, *fip37*L, and *nerd-1* plants. *P < 0.05; ***P < 0.001; *t* test. **(F)** ITN GENE1 and methylated control mRNAs are m⁶A-methylated in an FIP37- and NERD-dependent manner. m⁶A-IP-qPCR analysis was performed on several ITN and methylated control loci in WT, *nerd-1*, and *fip37*L seedlings. Error bars, mean ± SD; n = 3 biological replicates. **(G)** Scatter plot representation of m⁶A enrichment score (on the Y-axis) and FC in downstream reads between WT and *fip37*L conditions (X-axis).

insight into the potential mechanism linking NERD to FIP37-dependent m⁶A regulation in plants, we assessed the impact of *nerd-1* mutation on m⁶A regulation by performing quantitative measurement of m⁶A/A ratio on mRNA using liquid chromatography-tandem MS (LC-MS/MS). As a control in this experiment, we also quantified m⁶A/A ratio in mRNAs extracted from *fip37*L. We determined the relative level of m⁶A/A in WT mRNA to be ~2.3%, which is comparable with the published m⁶A/A data in WT *Arabidopsis* (Fig 3E) (Shen et al, 2016; Růžička et al, 2017). Interestingly, we found that the global m⁶A/A levels in mRNA was reduced by 25% in *nerd-1*–mutant background (Fig 3E), indicating that NERD impacts mRNA m⁶A dynamics in plants, although to a lesser extent than FIP37 (Fig 3E). To extend this analysis, we surveyed *A. thaliana* m⁶A methylome data

(Luo et al, 2014; Shen et al, 2016; Anderson et al, 2018) and found that a majority (65%) of ITN GENE1 mRNAs contained a high-confidence m⁶A peak near the 3′-UTR region (Table S2). Retrieval and analysis of FIP37 transcriptome-wide m⁶A methylome data (Shen et al, 2016), further indicated that ITN GENE1 mRNAs are methylated in an FIP37-dependent manner (Fig S8A), an observation that prompted us to assess the impact of *nerd-1* mutation on m⁶A methylation at those loci. By performing m⁶A-IP-qPCR assays on ITN GENE1s and control loci previously shown to be methylated in an FIP37-dependent manner (Fig S8B) (Shen et al, 2016), we confirmed that ITN GENE1 transcripts were m⁶A-tagged in an FIP37-dependent manner (Fig 3F). More importantly, NERD deficiency led also to a decrease, although to a lower extent, of m⁶A levels at all loci tested, including control loci

(Fig 3F), supporting the notion that NERD is required to reach full level of m⁶A modification on mRNA. However, no general correlation could be observed between the level of FIP37-dependent m⁶A enrichment and the control of readthrough transcription (Figs 3G and S8B and Table S6), suggesting that ITN loci require m⁶A modification to restrict chimeric mRNA formation in a qualitative manner.

### The m⁶A pathway restricts mRNA chimera formation at ITN loci by assisting GENE1 mRNA polyadenylation

To gain insight into the mechanism by which FIP37-dependent m⁶A pathway could restrict mRNA chimera formation at ITN loci, we performed rapid amplification of cDNA 3'-ends (3' RACE) on total RNA isolated from WT and *fip37*L plants. To do so, total RNAs were reverse-transcribed with an oligo (dT) anchor primer and amplified sequentially with nested primers located in ITN GENE1 sequence together with the anchor primer (Fig 4B). Variable patterns of polyA sites were detected at tested ITN GENE1 loci in WT plants (Fig 4A), including cases in which GENE1 have a prominent polyA site cluster (AT1G05890 and AT2G21440) and cases in which GENE1 shows more complex profiles of polyA sites (AT1G71340 and AT5G07630) (Fig 4A and B). The nature of polyA sites was confirmed by sequencing the corresponding PCR fragments, and, in some cases, by the examination of polyA tags (PATs) database obtained from high-throughput sequencing analysis in *A. thaliana* (Fig 4B) (Wu et al, 2011; Thomas et al, 2012). Our analysis revealed that the appearance

of long polyadenylated chimeric mRNA transcripts at ITN loci (noted as pAc and confirmed by sequencing) was concomitant with the reduced accumulation of polyadenylated GENE1 mRNA forms in the *fip37*L background (pA highlighted in red in Fig 4A and B). This effect was specific because the accumulation of polyadenylated mRNAs at control loci (AT2G23350 and AT2G18790) was not affected in *fip37*L (Fig 4A–C). Taken together, our results suggest that the m⁶A pathway restricts mRNA chimera formation at ITN chimeric loci by assisting polyadenylation of GENE1 mRNA.

### m⁶A-assisted polyadenylation of GENE1 mRNA requires the YTHDC-type domain-containing protein CPSF30L

The most prominent family of m⁶A-readers corresponds to the YT521-B homology (YTH) protein clade (Meyer & Jaffrey, 2017). A likely candidate gene for linking FIP37-dependent m⁶A modification to mRNA polyadenylation is the *CPSF30* gene because, in plants, it encodes two isoforms (CPSF30S and CPSF30L) of the conserved 30-kD subunit of cleavage and polyadenylation specificity factor complex (CPSF) that differ by the presence of an YT512-B Homology Domain-containing (YTHDC)-type protein domain in CPSF30L (Liu et al, 2014b; Shi & Manley, 2015; Wu et al, 2015; Berlivet et al, 2019) (Figs 5A and S9F). To assess the implication of CPSF30S and CPSF30L in the m⁶A-assisted polyadenylation of GENE1 mRNA, we characterized two T-DNA insertion lines in *CPSF30*, the first one lacking both CPSF30 isoforms (*cpsf30-1*; also known as *oxt6*) (Liu et al, 2014b;

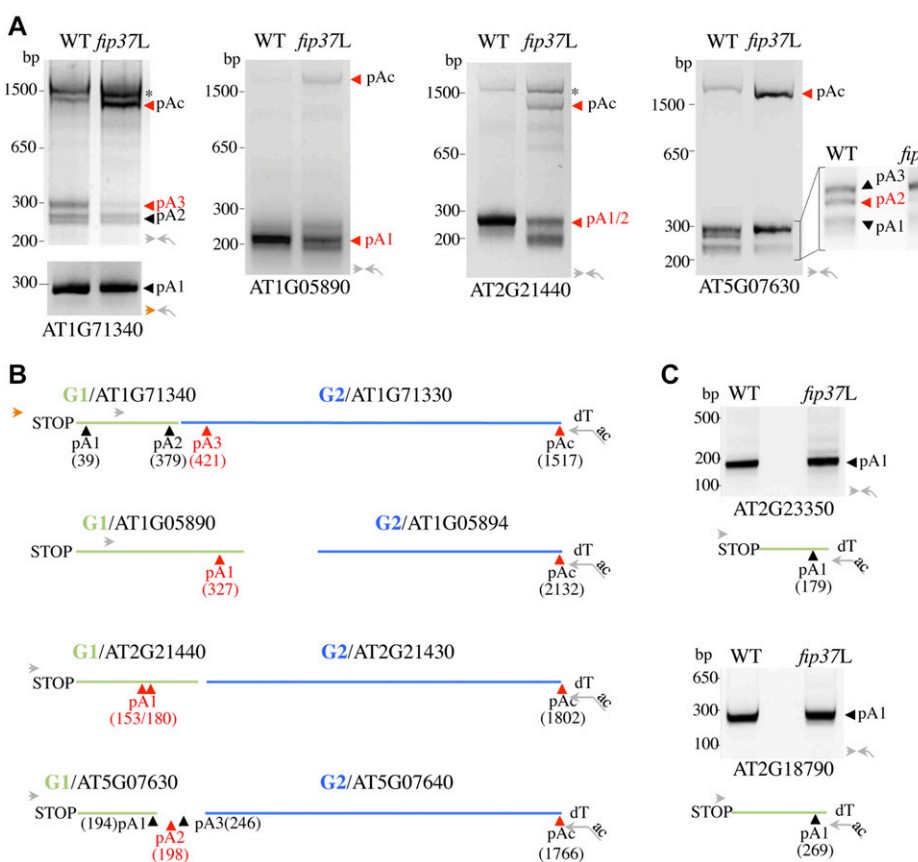

**Figure 4. m⁶A-assisted polyadenylation controls mRNA chimera formation at ITN loci.**
**(A)** 3' RACE gel images of ITN loci in WT and *fip37*L plants. RACE assays were performed with total RNA isolated from 9-d-old seedlings. The polyadenylated sites corresponding to the ITN GENE1 (pAx) are indicated at the bottom of the gel, whereas the polyadenylated sites corresponding to the mRNA chimera (pAc) migrate at the top. The FIP37-dependent GENE1 polyadenylation sites are marked in red. **(B)** The primers used in the 3' RACE are indicated with arrowheads and are localized on the corresponding sequences in (B). The oligo (dT) anchor primer is indicated with a broken arrow. **(B)** PolyA sites detected at ITN loci. The position of the polyA sites is indicated at the X-axis relative from ITN GENE1 STOP codon. The respective positions of ITN GENE1 and GENE2 loci are indicated with green and blue lines. Primers used in RACE assays are indicated with arrowheads and broken arrow. **(C)** 3' RACE gel images of m⁶A methylated control loci in WT and *fip37*L plants. RACE assays were performed with total RNA isolated from 9-d-old seedlings. The position of the polyA sites is indicated at the X-axis relative from ITN GENE1 STOP codon.

Delaney et al, 2006), and a novel one that specifically abrogates *CPSF30L* mRNA production (*cpsf30-3*) (Figs 5A and S9A). We then performed RNA-seq analysis on purified mRNA from WT, *cpsf30-1*, *cpsf30-3*, and *fip37*L lines and calculated the changes in read density in the 500-bp windows downstream of annotated genes in these plant backgrounds (Table S5). This analysis revealed a significant (*P*-value < 0.0001; hypergeometric test) overlap between genes showing readthrough transcription in *fip37*L and *cpsf30-1* (142 out of 174 i.e., about 81% of FIP37-dependent loci; Fig 5B and Table S5),

indicating that CPSF30-dependent activity is likely engaged in the control of readthrough transcription at FIP37-dependent loci. This conclusion was confirmed at ITN loci as exemplified by Integrative Genomics Viewer snapshots and using semiquantitative RT–PCR analysis (Fig S9B and D). However, in agreement with previous analysis (Thomas et al, 2012), *cpsf30-1* mutant exhibits a broader impact on 3′-end formation, defining a large set of 618 loci showing no dependency on FIP37 for readthrough transcription and polyadenylation controls, hereafter named as FIP37-independent

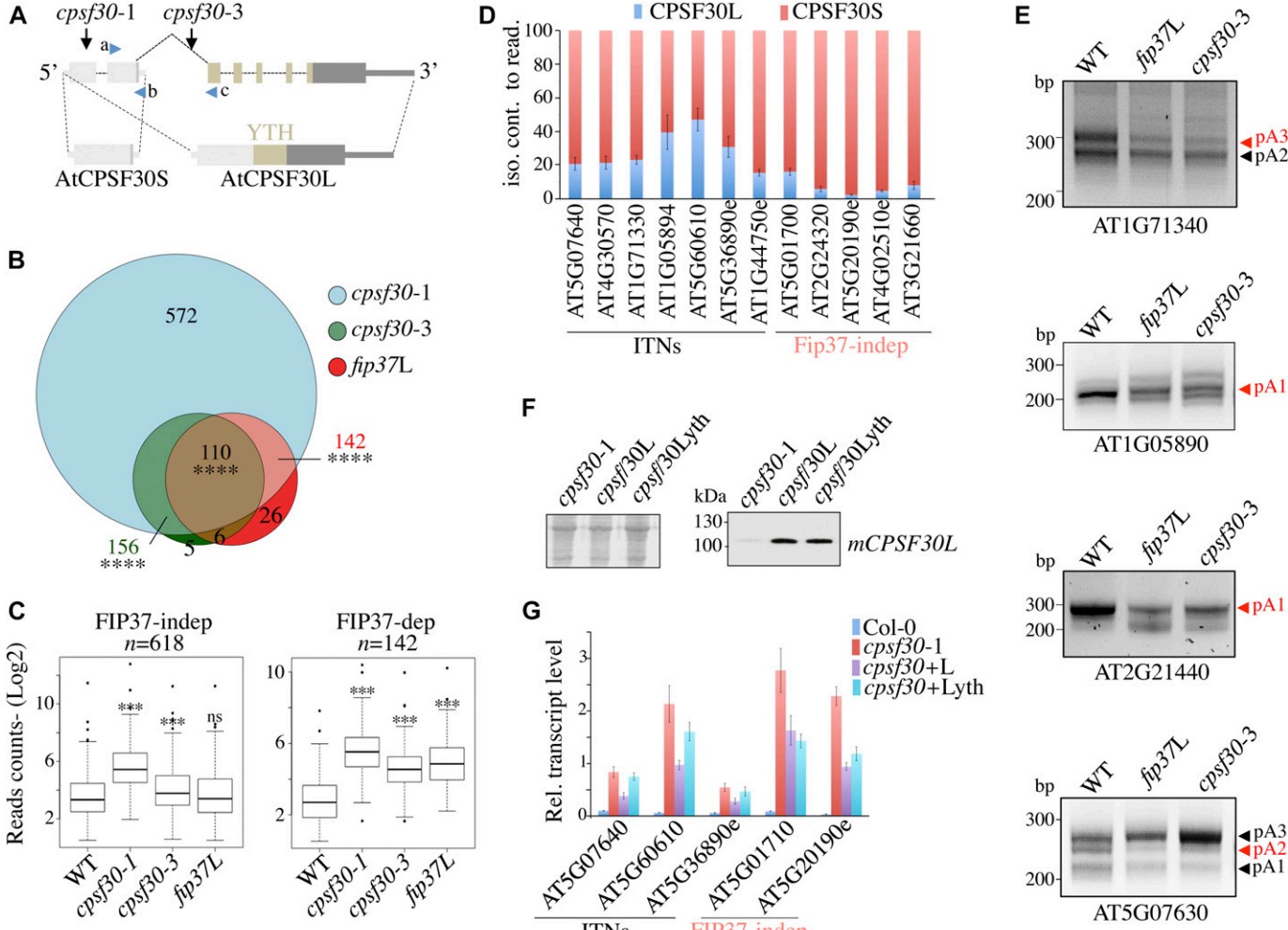

**Figure 5.  CPSF30L links m$^6$A methylation to site-specific polyadenylation at ITN loci.**
**(A)** Structure of the *CPSF30* gene and mRNA transcripts. Exons and UTRs are shown with thick and thin grey bars, respectively. Introns are shown with black-dashed lines. APA at Intron-2 produces *CPSF30S* mRNA, whereas alternative splicing of Intron-2 generates *CPSF30L* mRNA. Location of the YTH domain in CPSF30L is indicated with a brown rectangle. The positions of T-DNA insertions in the *cpsf30* mutants are indicated. Primers used in the qRT–PCR experiments are indicated. **(B)** Overlap between loci showing a significant increase in downstream read counts in *cpsf30-1*, *cpsf30-3*, and *fip37*L plants. **** denotes *P*-value < 0.0001 for enrichment in the overlaps, hypergeometric statistical test. **(C)** Boxplot showing the read counts in the 500-bp downstream region for FIP37-dependent and FIP37-independent CPSF30-responsive loci in WT, *cpsf30-1*, *cpsf30-3*, and *fip37*L plants. ns, not significant; ***P < 0.001; *t* test. **(D)** Relative contribution of CPSF30S and CPSF30L isoforms to mRNA chimera formation control at ITN- and FIP37-independent CPSF30-responsive loci. Data were calculated from the qRT–PCR analysis of ITN- and FIP37-independent CPSF30 chimeric/extended transcripts presented in Fig S9E. Data represent the means of three independent experiments and error bars corresponds to SD values. **(E)** Iso. Cont. to read. stands for isoform contribution to readthrough and is expressed in % (E) 3′ RACE gel images of ITN loci in WT, *fip37*L, and *cpsf30-3* 9-d-old seedlings. RACE assays were performed as previously described. **(F)** Western blot analyses of *cpsf30-1* mutant plants expressing Myc-tagged WT (30L) and *yth* mutant (30Lyth) versions of CPSF30L. The corresponding Coomassie staining is represented on the left part. The anti-Myc antibody was used to visualize the tagged proteins (right part). **(G)** qRT–PCR analysis of ITN- and FIP37-independent CPSF30-responsive mRNAs accumulation in WT, *cpsf30-1*, *cpsf30-1*+30L, and *cpsf30-1*+30Lyth plants. Relative transcript levels were normalized to actin using the ΔΔCt method. Data represent the means of four independent experiments and error bars correspond to SD values.

CPSF30-sensitive loci (Figs 5B and C, S9C, and D). Interestingly, by comparing the set of genes showing readthrough transcription in *cpsf30-3*, we found that the corresponding loci were significantly (*P*-value < 0.0001; hypergeometric test) included within the set of *cpsf30-1*–dependent ones (156 out of 167 i.e., about 93% of *cpsf30-3*–dependent loci; Fig 5B), as expected from the common genic origin of those mutants. More importantly, we also found a significant overlap (*P*-value < 0.0001; hypergeometric test) between *cpsf30-3*– and *fip37*L–dependent loci (110 loci i.e., about 70% of *cpsf30-3*– and 66% of *fip37*L-dependent loci; Fig 5B and Table S5), supporting the idea of a functional convergence between these proteins. This observation was further supported by boxplot and real-time RT–PCR analyses, reporting a more significant contribution of CPSF30L to readthrough control at FIP37-dependent versus FIP37-independent CPSF30-sensitive loci (Figs 5C and D, and S9E). To extend this analysis, we conducted 3′ RACE procedure on total RNA extracted from *fip37*L and *cpsf30-3* seedlings and observed that these mutations affected polyA sites patterns similarly at ITNs (Fig 5E). To assess the contribution of the YTHDC-type domain in CPSF30L activity, we transformed either a WT (30L) or an YTHDC-mutated version of CPSF30L (30Lyth), altered in the conserved m6A-binding aromatic cage (see W258, W309, and Y318 in Fig S9F) in *cpsf30-1* mutant background, and selected transgenic lines expressing closely similar levels of these proteins (Fig 5F). We then tested the ability of these CPSF30L variants to complement readthrough transcription defects at ITN- and FIP37-independent CPSF30-sensitive loci. Although CPSF30L and CPSF30Lyth showed a similar and partial ability to complement readthrough transcription at FIP37-independent CPSF30-sensitive loci, they differed in their ability to restrict chimeric mRNA formation at ITN loci (Fig

5G), suggesting that the YTHDC-type domain of CPSF30L is essential to control chimeric mRNA formation at ITN loci.

## m6A-assisted polyadenylation restricts chimeric gene-TE transcript formation

During our analysis of ITN loci, we noticed two cases in which GENE2 correspond to a mis-annotated TE (Table S3), suggesting that m6A-assisted polyadenylation could also exert an inhibitory control over chimeric GENE-TE transcripts. To extend this observation, we focused our analysis on the AT3G47890-AT3G47875 candidate locus, in which the mis-annotated GENE2 corresponds to a non-LTR retrotransposon of the LINE (long interspersed nuclear element, AT3TE71565) family (Table S3). Sequence comparison in *A. thaliana* and *A. lyrata* reveals evolutionary conserved regions flanking AT3TE71565 indicating that the LINE1 element got inserted within AT3G47890 3′-UTR posterior to the separation between the *A. thaliana* and *A. lyrata* lineages (Figs 6A and S10D). As such, AT3TE71565 contains all of the typical hallmarks of LINE1-mediated retrotransposition, including the presence of target-site duplications, 5′-truncated ORFs, and a short polyA tail confirming that AT3TE71565 is inserted in a tail-to-head orientation relative to the AT3G47890 gene (Figs 6A and S10A). Visual inspection of the mRNA-seq reads and RT–PCR assays further confirmed the accumulation of chimeric AT3G47890-AT3TE71565 transcripts in *fip37*L- and *cpsf30*-mutant backgrounds (Figs 6B and S10C). 3′ RACE analysis further revealed that the appearance of polyadenylated chimeric RNA transcripts in both fip37L- and *cpsf30-3*–mutant backgrounds was concomitant with the reduced accumulation of a long poly-adenylated GENE1 mRNA form (pA2 highlighted in red in Figs 4A and B,

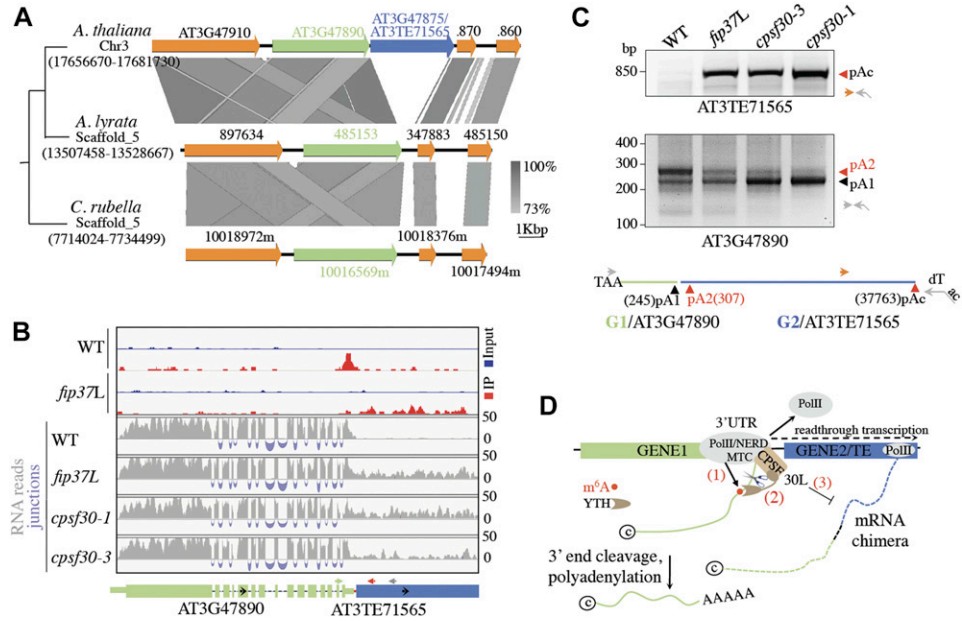

**Figure 6. m6A-assisted polyadenylation restricts chimeric GENE-TE transcript formation in *Arabidopsis*.**
**(A)** An example of a LINE1-type retroelement translocation that has occurred recently in *A. thaliana* (top), after the separation from the *A. lyrata* (middle) and *C. rubella* (bottom) lineages. **(B)** Normalized reads mapping to the chimeric AT3G47890-AT3TE71565 loci are shown in WT, *fip37*L, *cpsf30-1*, and *cpsf30-3* backgrounds. Upper panel shows m6A peaks as revealed by m6A-seq performed in WT and *fip37*L backgrounds (Shen et al, 2016). Light blue color represents input reads, whereas red color represents IP reads. The transcript structures and annotations are the same as in Fig 1C. Primers used in the RT–PCR and McrBC experiments are shown as green, red, and grey arrowheads, respectively. **(C)** 3′ RACE gel images of chimeric AT3G47890-AT3TE71565 loci in WT, *fip37*L, *cpsf30-3*, and *cpsf30-1* plants (top panel). RACE assays were performed as previously described. pA2 corresponds to the AT3G47890 polyadenylated site that decreases in *fip37*L and *cpsf30-3* plants, whereas pAc indicates the polyadenylated sites corresponding to the mRNA chimera. PolyA sites detected at AT3G47890/AT3TE71565 ITN loci are indicated

relative to GENE1 STOP codon (bottom panel). The respective positions of ITN GENE1 and TE loci are indicated with green and blue lines. The primers used in the 3′ RACE are indicated with arrowheads. The oligo (dT) anchor primer is indicated with a broken arrow. **(D)** Model for m6A-assisted polyadenylation (m-ASP) at ITN loci. NERD and FIP37 are required to reach full m6A deposition at all loci including ITN GENE1s (1). The recognition of m6A by the YTHDC-type domain of CPSF30L promotes cleavage and polyadenylation at GENE1 3′-UTR (2), therefore restricting mRNA chimera formation (3).

and S10D), supporting the idea that m⁶A-assisted polyadenylation also restricts chimeric GENE-TE transcript formation in plants. Using McrBC-PCR analysis, we found no change in AT3TE71565 DNA methylation levels in *fip37*L and *cpsf30-1/3* mutants (Fig S10B and E), indicating that RNA chimera formation likely accounts for the up-regulation of RNA signal at AT3TE71565. Taken together, our results show that m⁶A-assisted polyadenylation can also restrict the formation of specific chimeric gene-TE transcript in plants.

## Discussion

Our study provides novel insights into the role of m⁶A pathway in plants, revealing the implication of plant-specific and conserved m⁶A-related proteins in polyadenylation and transcription termination control at rearranged genomic loci in *A. thaliana*. The first evidence that m⁶A could participate in the control of chimeric mRNA formation at specific loci in plants came from our ongoing analysis of NERD, a plant homeodomain finger- and AGO hook–containing protein that, until now, was thought to contribute to the transcriptional silencing of newly acquired genomic sequences via a non-canonical siRNA-dependent DNA methylation pathway (Garcia et al, 2012; Pontier et al, 2012). Surprisingly, our studies point to an unexpected activity in that NERD depletion reduces mRNA m⁶A abundance at a genome-wide level, resulting in transcription termination defect and chimeric mRNA formation at rearranged genomic loci. We show that the contribution of NERD in mRNA chimera control is independent of its AGO hook–binding platform, indicating that NERD is likely to achieve multiple functions in plants.

Our finding that the NERD depletion decreases m⁶A deposition on mRNAs provides a mechanistic basis for NERD-mediated regulation of chimeric mRNA formation at ITNs; however, it does not explain the exact role of NERD in m⁶A regulation in plants. Despite our epistatic genetic interaction data, all attempts made to detect a physical interaction between NERD and FIP37 in vivo failed so far, suggesting that NERD is not likely to be a novel core component of m⁶A methyltransferase complex. However, recently published Arabidopsis proteomics data indicate that NERD copurifies with SPT4/SPT5 transcription elongation factors and elongating Pol II in vivo (Antosz et al, 2017). In addition, NERD also harbors a specific protein domain, named as Plus3 (Pontier et al, 2012), which has been shown to mediate Pol II-associated factor 1 complex (PaF1C) recruitment to genes by binding to a specific domain within the Pol II elongation factor SPT5 (Mayekar et al, 2013; Wier et al, 2013). This suggests that NERD could potentially act at an early stage of m⁶A deposition within the elongating RNA polymerase II (Pol II) complex. Together with our previous data showing that NERD binds unmethylated histone H3 lysine 4 (H3K4me0) through a conserved plant homeodomain finger-type domain (Pontier et al, 2012), these observations suggest that NERD could act at the interface between elongating Pol II and chromatin, potentially facilitating the co-transcriptional deposition of m⁶A on mRNAs, in agreement with the prevailing model for m⁶A deposition (Ke et al, 2017).

Our study revealed an unexpected function of m⁶A pathway in the control of transcriptome integrity by guiding site-specific mRNA

polyadenylation at a subset of genomic loci that exhibit intrinsic transcription termination and polyadenylation defects (Fig 6D). The reported m⁶A-assisted polyadenylation (m-ASP) pathway is distinct from the classical mechanisms of APA (Wu et al, 2011; Xing & Li, 2011): it relies on plant-specific and conserved m⁶A-related proteins, including a dedicated m⁶A reader, and it targets rearranged genomic loci that are prone to produce chimeric RNAs (Fig 6D). Taken together, our results, therefore, point to a selective recognition of 3′-UTR m⁶A as a safeguard mechanism that emerged in plants to restrict inappropriate gene expression and ensure transcriptome integrity.

Although the implication of m⁶A in mRNA stability and translation has been well documented (Meyer & Jaffrey, 2017), the role of m⁶A modification in the control of polyadenylation site selection remains more obscure, despite the prevalence of m⁶A within 3′-UTRs. In animals, m⁶A has been shown to mediate opposing effects on site selection at genes experiencing APA, favoring either proximal or distal polyadenylation depending on the cell type (Ke et al, 2015; Yue et al, 2018). The recent identification of YTHDC1 or VIRMA as components of the m⁶A pathway that interact with 3′-end processing factors and modulate APA in animals provides a molecular basis to explain these effects (Kasowitz et al, 2018; Yue et al, 2018). A direct role for m⁶A in targeting mRNA polyadenylation in plants is supported by our finding that the concomitant depletion of FIP37 and m⁶A suppresses specific polyA sites at a subset of genomic loci, in fine leading to transcription readthrough, intergenic splicing, and chimeric mRNA formation. Given that chimeric mRNA formation in plants can also be controlled by m⁶A-independent mechanisms, as demonstrated at FPA-only loci, our study raises the possibility that the selective recruitment of m-ASP pathway to sustain efficient transcription termination and polyadenylation may depend on the local genomic context. Such hypothesis is supported by our observation that ITN loci are enriched in recently rearranged gene pairs and display an atypical chromatin signature, with GENE1 harboring canonical histone marks of both active transcription initiation and elongation (i.e., H3K4me3 and H3K36me3/H2Bub, respectively), and GENE2 solely exhibiting marks of transcription elongation. We propose that the presence of a dual peak of transcription elongation marks over ITN gene pairs likely reflects the basal level of pervasive readthrough transcription that characterizes these loci in WT plants. Whether the atypical state of chromatin and the presence of m⁶A at ITNs shape the transcription termination and polyadenylation responses to pervasive readthrough transcription remains to be clarified. Interestingly, evidence for cross-talk between histone modification and RNA methylation machineries has recently emerged in animals (Huang et al, 2019), reinforcing the idea that chromatin- and m⁶A-related pathways can influence one another.

Our study demonstrates that m⁶A-assisted polyadenylation at ITN loci requires CPSF30L, a poorly characterized YTHDC-type domain-containing isoform of the *A. thaliana* gene *CPSF30* (Fig 6D) (Berlivet et al, 2019). Although several studies have previously found a link between *CPSF30* gene and APA in plants, this activity has been mostly attributed to CPSF30S, the YTHDC-less short isoform, previously shown to be sufficient to restore developmental and molecular phenotypes incurred by mutations in *CPSF30* (Liu et al, 2014b; Shi & Manley, 2015; Wu et al, 2015). The role

of CPSF30L could have been underappreciated in previous studies because they were focused on exploring gene-specific APA site distribution (Thomas et al, 2012). Our study also pinpoints the concerted action of both CPSF30 isoforms at ITN loci, indicating that both m⁶A-dependent and m⁶A-independent polyadenylation mechanisms contribute to transcription termination at those loci. Such cooperation could be promoted by the presence of a conserved N-terminal zinc-finger domain shared by both CPSF30 isoforms that is believed to self-associate (Delaney et al, 2006). Concerning the CPSF30L isoform, our data show that its role in chimeric mRNA formation control is contingent on FIP37 activity and on the integrity of the conserved aromatic cage known to bind m⁶A in other YTH proteins (Wu et al, 2017). These equivalence indicate that the regulatory function of m⁶A require CPSF30L, and that CPSF30L is likely an m⁶A reader whose recruitment is needed to ensure optimal or near-optimal transcription termination and polyadenylation at ITN loci (Fig 6D).

Finally, our results suggest that the m-ASP pathway can also restrict chimeric GENE-TE transcripts produced upon the insertion of TE in the 3′-untranslated region of genes. Although such events are rare, chimeric gene-TE transcripts are a mean by which retrotransposons can mobilize or lead to gene retroposition in plants and animals, potentially interfering with genome function (Kazazian, 2004; Xiao et al, 2008; Eickbush & Eickbush, 2011; Zhu et al, 2016). Although specific DNA methylation-based silencing mechanisms have been developed by plants to restrict TE expression per se (Law & Jacobsen, 2010), our data suggest that the m-ASP pathway may act, at a locus-specific level, as an additional mechanism to limit the inappropriate expression of TE inserted in the vicinity of genes.

# Materiel and Methods

## Plant methods

*A. thaliana* Col-0 as well as the following mutants were used in this study: *nerd-1* (SALK_093814) (Pontier et al, 2012), *nerd-3* (GABI_123G03), *fca-9* (Liu et al, 2007), *fpa-7* (SALK_138449), *cpsf30-*1 (Liu et al, 2014b), *cpsf30-3* (GABI_477H04, obtained from the Nottingham Arabidopsis Stock), *fip37-4 LEC1:*FIP37 (Shen et al, 2016). The primers used to genotype new lines are listed in Table S7. *nerd-1*+T corresponds to the complemented *nerd*-1 line described in (Pontier et al, 2012). The same construct was introduced into *nerd-3* plants by crosses to obtain *nerd-3+T*. For the epistasis analysis, *fip37-4* LEC1:FIP37 heterozygous plants were crossed to *nerd-1* mutants. The progenies were screened by PCR (see Table S7) to retrieve WT LEC1: FIP37, *nerd-1* LEC1:FIP37, and *nerd-1 fip37-4* LEC1:FIP37 lines.

Plants were either grown in soil or cultivated in vitro on plates containing 2.20 g/l synthetic Murashige and Skoog (MS) (Duchefa) medium, pH 5.7, and 8 g/l agar. For transgenic seeds selection, 30 μg/l hygromycin were added. For *fip37-4* mutants or derivatives, experiments were carried out on plates containing 4.41 g/l MS medium, pH 5.7, 1% sucrose, and 8 g/l agar; 9-d-old seedlings were collected under a microscope. In growth chambers, the conditions are 20°C, 60–75% hygrometry with a 16-h light/8-h dark photoperiod

(100 μE m⁻² s⁻¹ light [fluorescent bulbs with white 6500K spectrum, purchased from Sylvania]). For in vitro culture, the seeds were surface-sterilized, sown on plates, incubated for 48 h at 6°C in the dark, and placed in a growth cabinet at 20°C with a 16-h-d/8-h-dark cycle and 130 μE m⁻² s⁻¹ light (LEDs with white 4500K spectrum, purchased from Vegeled).

## Generation of transgenic plants

To generate AT4G30580/70m and AT1G71340/30m reporters, the corresponding genomic sequences were amplified using TL3437(KpnI)-TL3436(BamHI) and TL3438(KpnI)-TL3439(BamHI) primer pairs, @ respectively. After sequencing, the corresponding fragments were introduced in a binary vector (CTL639) containing a 4cMyc tag in front of a NosT terminator and the hygromycin resistance gene driven by a *35S* promoter (CTL235). The obtained plasmids were used to transform *nerd-1* plants via Agrobacterium transformation.

To obtain WT (30L) and YTHDC-mutated version of CPSF30L (30Lyth) constructs, we amplified the *CPSF30* promoter using TL3400 and TL3401. We also gene-synthetized (Genecust) the cDNAs corresponding to the WT and the YTHDC-mutated versions of *CPSF30L* (W258 to A, W309 to A, Y314 to A mutations; Fig S9F) with a 4cMyc tag upstream the ATG. Both fragments were introduced upstream of a NosT terminator into a binary vector containing the hygromycin resistance gene driven by a 35S promoter (CTL235). The obtained plasmids were used to transform *cpsf30-1* plants via Agrobacterium transformation.

## RNA sequencing

Total RNA was extracted from 14-d-old seedlings for Col0, *nerd-1*, and *nerd-1+T* plants and from 9-d-old seedlings for Col-0, *nerd-1*, and *cpsf30-1* plants using the RNeasy Plant Mini Kit (Cat. No. 74904; QIAGEN) and treated onto the column with the RNase-Free DNase Set (Cat. No. 79254; QIAGEN). Two replicates were performed per line. PolyA purification, library preparation using the Illumina TruSeq Stranded mRNA Kit and library quality controls were performed by Fasteris SA. For Col-0, *nerd-1*, and *cpsf30-1*, sequencing was performed by Fasteris SA with paired reads, 2 × 125 pb and 37–53 × 10⁶ reads per library. For Col-0, *nerd-1*, and *nerd-1-T,* sequencing was performed by Fasteris SA with single-end reads, 1 × 125 bp and 21–52 × 10⁶ reads per library.

## RNA-seq analysis

For each library, 20–30 × 10⁶ reads were obtained with 85–90% of the bases displaying a Q-score ≥ 30, with a mean Q-score of 38 as assessed with FastQC (http://www.bioinformatics.babraham.ac.uk/projects/fastqc/). Filtering out of reads corresponding to chloroplastic, mitochondrial, and rRNA genes was performed with Bowtie 2 v2.3.4 in sensitive local mode (Langmead & Salzberg, 2012). Remaining reads were mapped against the TAIR10 genome using gtf annotation file and default parameters of TopHat2 v2.0.7 (Kim et al, 2013). Assembly and transcript quantification were performed with Cufflinks v2.2.1 (Trapnell et al, 2010). Finally, the low-expressed transcripts, less than 1 RPKM (reads per kilobase per million

mapped reads), in one of the libraries were filtered out. The differential analysis was conducted by Cuffdiff software, belonging to the Trapnell suite. Quantification over 500-pb windows was made with BEDTools software (Quinlan & Hall, 2010). The 500-pb downstream from gene windows was defined using gtf annotation from the TAIR10 database. We normalized raw read counts by the total of mapped reads (rpm). Fold change (FC) was determined as the ratio between the normalized read counts between mutant and wild type. The list was further filtered to remove readthrough cases where the expression of the upstream genes changes in the *nerd-1* mutant background.

### Expression analysis

Total RNA was isolated from 50 mg of seedlings using the TRI reagent (Cat. No. TR-118; Euromedex) according to the manufacturer's instructions. Genomic DNA was then digested out using the RQ1 RNase-Free DNase (Cat. No. M6101; Promega): 4 $\mu$g of RNA was treated with 4 units of the enzyme in the 1× reaction buffer in a 20-$\mu$l total volume at 37°C for 30 min and the reaction stopped by adding 2 $\mu$l of RQ1 DNase Stop Solution and incubating 10 min at 65°C. cDNAs were obtained from 400 ng of RQ1-treated RNA using 1 $\mu$l of GoScript Reverse Transcriptase (Cat. No. A5003; Promega) in a 20-$\mu$l final volume reaction using random primers (Cat. No. C1181; Promega) in the presence of 20 units of RNasin (Recombinant Ribonuclease Inhibitor, Cat. No. C2511; Promega) and dNTP mix at a final concentration of 0.5 mM of each dNTP (Cat. No. U1511; Promega). Semi-quantitative RT–PCR amplifications were performed on 1 $\mu$l of cDNA in a 12.5-$\mu$l reaction volume to start with. The amplification of EF1-4$\alpha$ transcripts (TL786-TL787, Table S7) was used to equilibrate. qRT–PCR was performed on a Light Cycler 480 II machine (Roche Diagnostics) by using the Takyon No ROX SYBR MasterMix blue dTTP kit (Cat. No. UF-NSMT-B0701; Eurogentec). Each amplification reaction was set up in a 10-$\mu$l reaction containing each primer at 300 nM and 1 $\mu$l of RT template in the case of total RNA with a thermal profile of 95°C for 10 min and 40 amplification cycles of 95°C, 15 s; 60°C, 60 s. Relative transcript accumulation was calculated using the ΔΔCt methodology, using *ACTIN2* as internal control (Livak & Schmittgen, 2001). Average ΔΔCt represents three experimental replicates with standard errors. Primers used are listed in Table S1.

PolyA+ RNAs were purified from RNA obtained by the TRI Reagent (Cat. No. TR-118; Euromedex) extraction from 100 mg of tissue, using the PolyATtract mRNA Isolation System III (Cat. No. Z5310; Promega) according to the manufacturer's instructions. cDNAs were then synthetized according to the above protocol using 50 ng of polyA+ RNA as starting material. Semiquantitative and quantitative RT–PCRs were performed as described above with a 1/25 dilution of cDNAs from polyA+ RNA as template for qRT–PCR.

### Northern blots

RNA was isolated from 14-d-old seedlings according to the protocol described by Duc et al (2013). 2–3 g of grinded material was resuspended in 7.5 ml of NTES buffer (100 mM NaCl, 10 mM Tris–HCl, pH 7.5, 1 mM EDTA, pH 8, and 1% SDS). After addition of phenol/chloroform/isoamyl alcohol (25:24:1) and incubation of 10 min at

room temperature, the samples were centrifugated (6,000$g$, 10 min, 4°C). Ethanol precipitation was then performed on the supernatants (around 6 ml) by adding 15 ml of 100% ethanol and 600 $\mu$l of 3 M Na acetate, pH 6. After 2-h incubation at −20°C and centrifugation (15 min, 6,000$g$, 4°C), the pellets were resuspended in 2 ml of water and precipitated with an equal volume of 4 M LiAc, at 4°C, overnight. After a 15-min centrifugation at 4°C and 16,000$g$ in a microfuge, the pellets were resuspended in 360 $\mu$l of water, incubated on ice for 15 min, and ethanol-precipitated in the presence of Na acetate, at −20°C overnight. After a 15-min centrifugation at 4°C and 16,000$g$ and a wash with 70% ethanol, total RNAs were resuspended in 200 $\mu$l of water and quantified using NanoDrop (Thermo Fisher Scientific). 500 $\mu$g to 1 mg of these RNAs were then used to obtain polyA+ RNA using the PolyATtract mRNA Isolation System III (Cat. No. Z5310; Promega) according to the manufacturer's instructions. Northern blots were performed using 2 $\mu$g of the obtained polyA+ RNA resuspended in 100% formamide. After a 5-min denaturation at 70°C, the RNAs were separated onto a 1% agarose, 0.5% formaldehyde gel in 0.03 M tricine 0.03 M triethanolamine buffer. The RNAs were transferred onto an NX-*Hybond* membrane (GE Healthcare) and cross-linked with EDC (1-ethyl-3-(3-dimethylaminopropyl)carbodiimide, Cat. No. E7750; Sigma-Aldrich). For the At4g30570-80 locus, the probe was PCR-amplified on genomic DNA using TL2980 and TL3205 primers (Table S7), purified using the GeneClean Turbo Kit (Cat. No. 111102400; MP Biomedicals), and labeled with the Prime-a-Gene Labeling System (Cat. No. U1100; Promega) according to the manufacturer's instructions. The hybridization was performed in the PerfectHyb Plus Hybridization Buffer (Cat. No. H7033; Sigma-Aldrich) at 68°C, overnight. For At1g71340, the TL3417 primer (Table S7) was labeled using the T4 polynucleotide kinase (Cat. No. M4101; Promega) and used as in a probe in the PerfectHyb Plus Hybridization Buffer (Cat. No. H7033; Sigma-Aldrich) at 42°C overnight. In both cases, the membranes were washed once in a 2× SSC, 0.1% SDS solution for 5 min at room temperature and twice in 0.5× SSC, 0.1% SDS solution for 15 min at 42°C or 68°C before exposure.

### RNA m⁶A quantification by targeted MS (LC-MRM)

Total RNAs from WT, *fip37*L, and *nerd-1* plants were isolated using TRIzol reagent (Euromedex) according to the manufacturer's instructions. Polyadenylated RNAs were extracted twice by oligo (dT) magnetic beads (Cat. No. Z5310; Promega), followed by removal of remaining rRNA with RiboMinus Plant Kit (A10838-08; Invitrogen). mRNA concentration was measured by NanoDrop and Qubit assays. 210 ng of mRNA were diluted in a total volume of 20 $\mu$l milliQ water. 3 $\mu$l of 0.1 M ammonium acetate, pH 5.3, and 0.001 U of Nuclease P1 (N8630; Sigma-Aldrich) were added. Incubation at 42°C for 2 h was performed. Then, 3 $\mu$l of 1 M ammonium acetate and 0.001 U of alkaline phosphatase (P4252; Sigma-Aldrich) were added. The mixture was incubated at 37°C for 2 h. Next, the nucleosides solution was diluted two times and was filtrated with 0.22 $\mu$m filters (Millex-GV, SLGVR04NL; Millipore). 1 $\mu$l of each sample was injected, and all samples were analyzed in triplicate using LC-MRM.

The nucleosides were then separated by 1290 LC systems (Agilent Technologies) using an Synergi Fusion-RP column (4 $\mu$m particle size, 250 mm × 2 mm, 80 Å) (00G-4424-B0; Phenomenex) at flow rates

of 0.4 ml/min. The mobile phase consisted of 5 mM ammonium acetate, adjusted to pH 5.3 with acetic acid (solvent A) and pure acetonitrile (solvent B). The 30-min elution gradient started with 100% phase A followed by a linear gradient to 8% solvent B at 13 min. Solvent B was increased further to 40% over 10 min. After 2 min, solvent B was decreased back to 0% at 25.5 min. Initial conditions were regenerated by rinsing with 100% solvent A for additional 4.5 min.

The detection was performed using Agilent TripleQuad 6490 in positive ion mode. The multiple reaction monitoring (MRM) transitions used for detection were m/z 268 to 136 for Adenosine (fragmentor voltage: 380 V, collision energy: 18 V, cell accelerator voltage: 1 V, and retention time: 13.1 min) and m/z 282.1 to 150.1 for m$^6$A (fragmentor voltage: 380 V, collision energy: 21 V, cell accelerator voltage: 1.5 V, and retention time: 16.5 min). The ESI source was set as follows: capillary tension: 2,000 V, nebulizer: 50 psi, gas flow rate: 15 L/min, gas temperature: 290°C, sheath gas flow rate: 12 L/min, and sheath gas temperature: 400°C. The MS was operated in dynamic MRM mode with a retention time window of 3 min and a maximum cycle time set at 800 ms. The peak areas were determined using Skyline 4.1 software and the ratio m$^6$A/A was calculated. Means, medians, standard errors, and SDs were calculated. Normality test, F-test, and t test analysis were performed on calculated ratio using MedCalc software.

### 3′ Rapid Amplification of cDNA ends (3′ RACE)

3′ RACE has been performed according to the protocol from Rodríguez-Cazorla et al, 2015 with slight modifications. Total RNAs were extracted from 100 mg of 9-d-old seedlings using the RNeasy Plant Mini Kit (Cat. No. 74904; QIAGEN) and treated onto the column with the RNase-Free DNase Set (Cat. No. 79254; QIAGEN). 5 μg of RNA was used to set up a 13-μl reverse transcription reaction with 2.88 μM of oligo (dT) anchored primer (TL3576, Table S7) and dNTP mix (0.6 mM final each). After 5 min at 65°C and 5 min at 4°C, the RT enzyme was added as follows: 4 μl of 5× SuperScript IV Reverse Transcriptase buffer, 1 μl of 100 mM DTT, 1 μl of SuperScript IV Reverse Transcriptase (Cat. No. 18090010; Invitrogen), and 40U of RNasin (Recombinant Ribonuclease Inhibitor, Cat. No. C2511; Promega). The reaction was performed at 60°C for 30 min and stopped at 80°C for 10 min. A first PCR reaction was performed in 50 μl total reaction with 5 μl of template using 0.3 μl GoTaq G2 DNA Polymerase (Cat. No. M7848; Promega), 0.2 mM each dNTP, 8 μl 10 μM PCR anchor primer (reverse primer, TL3577, Table S7), and 1 μL 10 μM specific primer (primer 1, Table S7). PCR was performed with a thermal profile of 94°C for 2 min; 35 cycles (94°C, 30 s, 57°C, 30 s, and 72°C, 2 to 4 min) and then 72°C, 10 min. The reaction was checked onto an agarose gel. The second amplification was performed on a 1-μl aliquot from the first PCR with the same conditions as for the first PCR except that 1 μl 10 μM PCR anchor primer (reverse primer, TL3577, Table S7) and 1 μL of a 10 μM nested specific primer (primer 2, Table S7) were used. The final products were separated on an UltraPure Agarose (Cat. No. 16500-500; Invitrogen) gel. Lower bands corresponding to Gene1 polyA sites in WT were chosen to be sequenced as well as upper bands that appeared in *fip37* samples that corresponded to readthroughs. To get better sequences for the readthroughs, the second PCR was performed using a primer closer

to Gene2 polyA sites (primer 3, Table S7). The DNA corresponding to bands to be sequenced was extracted using the GeneClean Turbo Kit (Cat. No. 111102400; MP Biomedicals) and subcloned into pGEM-T Easy Vector (Cat. No. A1360; Promega). The DNA from 48 to 96 independent clones was sequenced with the SupremRun Service from GATC Biotech. The sequences were aligned and the position of the last nucleotide upstream the polyA tail was identified, the starting point being the first base downstream the ITN GENE1 stop codon. The number of clones and their precise position are reported in Table S7.

### m$^6$A-IP-qPCR

m$^6$A-IP-qPCR was performed as previously described (Dominissini et al, 2012; Shen et al, 2016). Total RNA was extracted and fragmented into ~200-nucleotide-long fragments by RNA fragmentation buffer (Cat. No. AM8740; Ambion). Fragmented RNA was incubated with 0.5 mg/ml anti-m$^6$A antibody (Cat. No. 202-003; Synaptic Systems) in IP buffer (150 mM NaCl, 0.1% Igepal CA-630, and 10 mM Tris–HCl, pH 7.4) supplemented with RNasin Plus RNase inhibitor (Cat. No. C2511; Promega) for 2 h at 4°C, and subsequently, protein A/G Plus-Agarose (Cat. No. sc-2003; Santa Cruz) that was pre-bound with BSA was added and incubated at 4°C for an additional 2 h. After extensive washing with IP buffer, bound RNA was eluted from the beads by incubation with 6.7 mM N$^6$-methyladenosine (Cat. No. M2780; Sigma-Aldrich) in IP buffer and precipitated by ethanol. The input and immunoprecipitated RNAs were reverse-transcribed with random hexamers (Cat. No. N8080127; Invitrogen) using M-MLV Reverse Transcriptase (Cat. No. M1701; Promega). Relative enrichment of each gene was determined by quantitative real-time PCR and calculated first by normalizing the amount of a target cDNA fragment against that of *TUB2* as an internal control, and then by normalizing the value for immunoprecipitated sample against that for the input. The primers used for real-time PCR are listed in Table S7.

### Protein extraction and immunodetection

Total plant protein extracts (up to 100 mg) were ground in liquid nitrogen with 200 μl SDS–PAGE loading buffer. Coomassie staining was used to calibrate loadings. Proteins were separated on SDS/PAGE gels and blotted onto Immobilon-P PVDF membrane (Cat. No. IPVH00010; MerckMillipore). Protein blot analysis was performed using the Immobilon Western Chemiluminescent HRP Substrate (Cat. No. WBKLS0500; MerckMillipore). The antibodies used in this study were HRP-coupled FLAG-specific antibody (Cat. No. A8592; Sigma-Aldrich) and anti-UGPase antibody as a loading control (Cat. No. AS05 086; Agrisera) at 1/5,000 dilution and HRP-coupled Myc-specific antibody (Cat. No. A5598; Sigma-Aldrich) at 1/10,000.

### Chromatin Immunoprecipitation

2 g of inflorescences were homogenized in 25 ml of Honda buffer (0.44 M sucrose, 1.25% Ficoll, 2.5% dextran T40, 20 mM Hepes KOH, pH 7.4, 10 mM MgCl$_2$, 0.5% Triton, and 5 mM DTT) supplemented with cOmplete EDTA-free Protease Inhibitor Cocktail (Cat. No. 11873580001; Roche) was filtered onto two layers of Miracloth (Cat.

No. 475855; MerckMillipore). 1% formaldehyde was added, and the samples were rotated 15 min at 4°C. The cross-link was stopped by adding glycine (0.125 N final) and rotating for 10 min at 4°C. Following a 15-min centrifugation (2,000*g*, 4°C), the nuclei were washed three times in Honda buffer, lysed in 1 ml of Nuclei lysis buffer (50 mM Tris–HCl, pH 8, 10 mM EDTA, 1% SDS, and cOmplete EDTA-free Protease Inhibitor Cocktail [Cat. No. 11873580001; Roche]) and sonicated using a Bioruptor (Diagenode). 20 *µ*g of chromatin was diluted and incubated with 5 *µ*l of anti-histone H3 antibody (Cat. No. 07-690; MerckMillipore) or anti-trimethyl-histone H3 (Lys4) antibody (Cat. No. 07-473; MerckMillipore) on a rotator overnight, at 4°C. 50 *µ*l of washed Dynabeads ProteinA/G (Cat. No. 10004D and 10002D; Invitrogen) were then added and incubated for 2 h at 4°C before the washes. After reversing the cross-link by a 10-min incubation at 95°C followed by a 1-h treatment with Proteinase K, the immunoprecipitated DNA was subjected to qPCR analysis using the primers listed in Table S7 and the Takyon No ROX SYBR Master Mix blue dTTP kit (Cat. No. UF-NSMT-B0701; Eurogentec) on a LightCycler 480 II machine (Roche Diagnostics).

### Analysis of DNA methylation

For McRBC analysis, DNA was extracted from 9-d-old seedlings using the DNeasy Plant Mini Kit (Cat. No. 69104; QIAGEN) and the concentration was measured using a NanoDrop (Thermo Fisher Scientific). 25 *µ*g of genomic DNA were digested with 15U of McrBC enzyme (Cat. No. M0272S; NEB) in a final volume of 50 *µ*l according to the manufacturer's instructions, at 37°C for 3 h, followed by a 15-min inactivation at 65°C. In parallel, the same mixture was prepared by replacing McrBC by H2O ("undigested sample"). The undigested/digested samples were analyzed by semiquantitative PCR using the primers indicated in Table S7.

### Gene duplication analysis

*A. thaliana*, *A. lyrata*, and *C. rubella* protein-coding genes were obtained from phytozome 11 (https://phytozome.jgi.doe.gov/pz/portal.html). MCScanX-transposed (Wang et al, 2013b) was used to classify *A. thaliana* genes into different gene duplication categories. Briefly, an intra- and inter-species all-against-all BLASTp of protein-coding genes was performed and the top five best no-self hits with an e-value lower than $1 \times 10^{-10}$ hit were retained for further analysis. Genes were assumed to be single when no matching sequence was found under the used BLASTp e-value threshold. Duplicated genes were classified into four categories: 1) whole genome duplication/segmental paralogs are located in corresponding collinear blocks within species. 2) Tandem duplicates correspond to paralogs that are immediately adjacent to each other. 3) Translocated duplicates refer to non-collinear paralogs that have at least one collinear ancestral paralog (Wang et al, 2013b). It should be noted that we prefer here the use of "translocated duplicates" rather than "transposed duplicates" used by the author of MCScanX-transposed because the latest term may be confused with transposition of TEs whereas gene movement and translocation may result from multiple mechanisms. 4) Other duplicates include duplicated genes that did not belong to any of the first three categories. Statistical tests were performed using R

and Figs 6A, S2B, and S6A were drawn using EasyFig software (Sullivan et al, 2011).

### GENE1/GENE2 evolutionary conservation and expression

To identify homologs of GENE1 and GENE2 in angiosperm and their genomic positions, the reference genome of 325 sequenced angiosperms was first downloaded from NCBI (ftp://ftp.ncbi.nlm.nih.gov/genomes/). The predicted protein sequences of each pair of genes were than blasted against the entire reference genome using the tblastn program, BLAST hits with a minimum of $1 \times 10^{-10}$ e-value corresponding to potential homologs sequences were all kept even if multiple alignments were found for each genes. Genes were considered as absent if no hit was found in the target genomes. When hit corresponding to both GENE1 and GENE2 were found, we consider genes as adjacent when any of the unique or multiple hits of GENE1 and GENE2 were located in the same chromosome/scaffold and their relative distance not greater than 5 kbp. Otherwise, GENE1 and GENE2 were classified as distant. For each GENE1/GENE2 pairs, the number of the four different scenarios shown in Fig 1C was calculated and expressed in percentage. The expression score in transcripts per million (TPM) was obtained from Araport11 (https://apps.araport.org/thalemine) and heat map generated using heatmap2 function in R.

### CS analysis of GENE1/GENE2 pairs

The genomic coordinates of CSs and of GENE1/GENE2 pairs were retrieved from published data (Sequeira-Mendes et al, 2014; Vergara & Gutierrez, 2017), and from the TAIR10 release of the *A. thaliana* genome. For each gene, the overlap with the nine CSs corresponding to proximal promoters (CS2), transcription start site (CS1), 5′ end of genes (CS3), long coding sequences (CS7), 3′ end of genes (CS6), polycomb chromatin (CS5), distal regulatory intergenic regions (CS4), AT-rich heterochromatin (CS8), and GC-rich heterochromatin (CS9) was determined. A Pearson's chi-square test was used to statistically assess the similarity between the distribution of GENE2s among the nine CSs for NERD- or FPA-dependent chimeric loci with that of all the genes immediately downstream of a gene associated with CS1 across the genome. Because of small sample size, *P*-values were computed by Monte Carlo simulation.

# Data Availability

Sequencing data have been deposited in NCBI' GEO under accession number GSE123912.

# Supplementary Information

# Acknowledgements

We thank Natacha Bies-Etheve, Mohamed-Ali Hakimi, Soizik Berlivet, Jean-Marc Deragon, Cecile Bousquet-Antonelli, and Olivier Panaud for discussions and comments on the manuscript, as well as Julia Buttin for help in CS distribution analysis and Véronique Suréda for help in genotyping. Lagrange laboratory research was supported by the Agence Nationale de la Recherche (grant 12-BSV6-0010), Centre National de la Recherche Scientifique, and Université de Perpignan Via Domitia. L. Shen laboratory work was supported by funding from National University of Singapore and Temasek Life Sciences Laboratory. F. Roudier laboratory work was supported by an attractiveness starting grant from ENS de Lyon. This study is set within the framework of the "Laboratoires d'Excellences (LABEX)" TULIP (ANR-10-LABX-41).

## Author Contributions

D Pontier: investigation, methodology and writing—original draft.
C Picart: investigation and methodology.
M El Baidouri: investigation, conceptualization, and writing—original draft.
F Roudier: investigation and writing—original draft.
T Xu: investigation.
S Lahmy: investigation and writing—original draft.
C Llauro: investigation.
J Azevedo: investigation and writing—original draft.
M Laudié: resources and investigation.
A Attina: investigation.
C Hirtz: methodology and writing—original draft.
M-C Carpentier: investigation, conceptualization, and writing—original draft.
L Shen: investigation and writing—original draft.
T Lagrange: conceptualization, supervision, funding acquisition, and writing—original draft.

## Conflict of Interest Statement

The authors declare that they have no conflict of interest.

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
