## [Reviewer comments · Life Science Alliance]

Life Science Alliance

The m6A pathway protects the transcriptome integrity by restricting RNA chimera formation in plants

Dominique Pontier, Claire Picart, Moaine ElBaidouri, François Roudier, Tao Xu, Sylvie Lahmy, Christel Llauro, Jacinthe Azevedo, Michèle Laudie, Aurore Attina, Christophe Hirtz, Marie-Christine Carpentier, Lisha Shen, , and Thierry Lagrange

DOI: <https://doi.org/10.26508/lsa.201900393>

Corresponding author(s): Thierry Lagrange, Centre National de la Recherche Scientifique/Université de Perpignan Via Domitia

Review Timeline:

Submission Date:	2019-04-01
Editorial Decision:	2019-04-18
Revision Received:	2019-05-14
Editorial Decision:	2019-05-17
Revision Received:	2019-05-20
Accepted:	2019-05-20

Scientific Editor: Andrea Leibfried

Transaction Report:

Please note that the manuscript was previously reviewed at another journal and the reports were taken into account in the decision-making process at Life Science Alliance.

Authors' response to previously raised concerns. Since the original reviewer reports cannot be displayed, questions/criticisms have been summarized by the editor.

Reviewer #1:

Specific points:

1. Is the overlap of upregulated genes in Fig 3A significant?

Response:

We agree with the reviewer's comment that the previous Fig.3A comparing the list of up-regulated loci from two different studies was confusing and not pertinent in the context of the paper. We have now executed a more stringent bioinformatic screen on *nerd* RNA-seq data, selecting for changes in strand-specific read density in the 500-bp windows downstream of all annotated genes upon NERD depletion and removing all candidate genes whose expression showed a significant change in *nerd* mutant background. This analysis revealed a more stringent population of NDT genes (n=107) that show a statistically significant downstream readthrough phenotype, but no changes in upstream gene expression (see Fig. 1A). Performing the same type of experiment on *fip37L* RNA-seq data, we also identified a set of 174 loci showing significant readthrough transcription in this mutant, and detected a significantly overlap with these two set of data (see Fig. 3C and supplemental Table S1 and S5), reinforcing the idea that FIP37 controls readthrough transcription at NDT loci.

2. Is there a mechanism explaining the specificity of NERD-dependent termination for only some genes? Why are m⁶A levels reduced in the *nerd-1* mutant background?

Response: We have provided novel data showing that NDT loci have a unique chromatin signature predictive of chimeric mRNA formation, with GENE1s having chromatin states characteristic of active transcription units while GENE2s are only associated with elongation marks (Fig. 2D and Supplemental Table S6). Such bias distribution of chromatin signatures is specific to NDT loci, and we therefore proposed that the atypical state of chromatin and the presence of m⁶A modification shape transcription termination and polyadenylation responses to pervasive readthrough transcription at NDTs.

Concerning the role of NERD in m⁶A regulation, all attempts made to detect a physical interaction between NERD and FIP37 *in vivo* failed so far, suggesting that NERD is not likely to be a novel core component of m⁶A methyltransferase complex. However, recently published proteomics data and NERD domain organization suggest that NERD could act at the interface between elongating Pol II and chromatin (see discussion), potentially facilitating the co-transcriptional deposition of m⁶A on mRNAs.

3. Does the loss of FPA also alter the level of m⁶A? Is there a specific enrichment of m⁶A peaks near the 3'UTR region in GENE1?

Response: We provide clear data in the paper showing that FPA-dependent chimeric loci differ from NDT-dependent loci in terms of gene expression and chromatin signature and are not targeted by m-ASP pathway (Fig. S4 and Fig 3B). We proposed that m-ASP pathway recruitment is related to the specific chromatin signature presented by NDT loci. It was not in our scope to extend further the analysis of FPA loci. Our current survey indicates that 63% of NDT GENE1 loci have high confidence m⁶A peak compared to roughly 20-30% of total

genes in Arabidopsis, supporting the idea that a majority of NDT loci are methylated in wild type plants.

4). The readthrough phenotype and presence or absence of m6A are correlated. Is there a causal link? Analysis of methylosome mutants or mutating m6A sites would provide such insight.

Response: We provided data indicating that the NDT loci are also reactivated in *mta/metl13* methyltransferase mutant in Arabidopsis (Supplemental Table S2), reinforcing the idea that the m6A methylation is required to restrict chimeric mRNA formation in plants. I would like to stress the point that the link between m6A and readthrough transcription control is also strongly reinforced by our data implicating CPSF30-YTH protein in the m-ASP pathway. I agree with the reviewer that mutating m6A sites would be very informative, but this type of experiments have hardly been performed in animal and are far to be feasible in plants. Indeed, we do not have CLIP data available in plants and such experiment will require to reintroduce a mutated version via plant transformation.

5) RNA-seq of the various mutants could show the impact of methylation-dependent versus -independent effects on readthrough transcription.

Response: We agree with the reviewer and have introduced RNAseq analysis of CPSF30S, CPSF30L and *fip37* that demonstrated that genes showing readthrough transcription in FIP37 mutant significantly overlap with the genes showing readthrough transcription in the plant missing the CPSF30-YTH protein (Fig. 5B,C). We also provide box-plot data reinforcing this claim, further reinforcing the claim that FIP37 and CPSF30L act in the same pathway.

Reviewer #2:

Specific points:

1. Does NERD indeed restrict chimeric RNA formation? Statistical analysis of variance is missing with only 2 RNA-seq replicates. Furthermore, RNA-seq data cannot easily allow identification of readthrough because up-regulation of the upstream gene can lead to increased detectability of reads downstream. Cases where the expression of the upstream gene changes should thus get excluded from the analysis (eg. psORF).

Response: We agree with the reviewer's comment. As indicated in the text, we applied a more stringent screen by selecting for changes in strand-specific read density in the 500-bp windows downstream of depletion, and removed all candidate genes whose expression showed a significant change in *nerd* mutant background. This analysis revealed a more stringent population of NDT genes (n=107) that show a statistically significant downstream readthrough phenotype, but no changes in upstream gene expression. These data are summarized in the box-plot data presented in Fig1 A and data presented in supplemental table S1).

We agree that the behavior of psORF could be misleading since this target, in contrast to other NDT loci, is dually controlled both by DNA methylation (Pontier et al., 2012) and m-ASP pathway (this work). This dual regulation makes the analysis of psORF more complicated, explaining why we omitted psORF in the list of NDT loci. However, we would like to keep the supplemental Fig 1, since the first evidence for a role of NERD in transcription readthrough control came from the analysis of psORF.

2. Clarifications needed for the sequencing and validation data. RNA-gel blot analyses supporting chimeric RNA formation is needed as well as an estimate of false positives. Sashimi plots annotating the number of splice junction events would be useful to assess the significance.

Response: We have introduced the position of the primers and provided sashimi plot for the NDT loci. We have also provided RNA gel-blot data on two representative loci that supported the formation of chimeric transcripts at NDT loci. As indicated in the text, we applied a more stringent screen by selecting for changes in strand-specific read density in the 500-bp window downstream of all annotated genes upon NERD depletion and removing all candidate genes whose expression showed a significant change in *nerd-1*. This analysis revealed a more stringent population of NDT genes (n=107) that show a statistically significant downstream readthrough phenotype but no change in upstream gene expression. These data are summarized in the box-plot data presented in Fig1 A and data presented in supplemental table S1).

3. Are Gene2 loci correctly annotated as genes? The loci may be alternative 3' ends of the upstream gene given that they lack 5'UTR annotation and are poorly expressed.

Response: We have included a phylogenetic study of NDT loci (excluding tandem genes which are repeats of the same gene), showing that GENE1 and GENE2 represent independent gene units and are not alternative 3' end version of GENE1 (Fig. 1B). This conclusion is further supported by our analysis showing that GENE2 are enriched in duplicated/translocated genes (see Fig. 2). Cryptic splicing events between GENE1 and GENE2 are often observed at NDT loci as indicated in Fig1B and supplemental Fig. S3.

4. The relationship of NERD to m6A is unclear, only few (58 of 3000) upregulated genes in *fip-37* overlap with those upregulated in *nerd* mutants.

Response: We have now introduced a figure based on Araport data summarizing that GENE2s at NDT loci are always silent in wild type plants (Fig. 1D), probably owing to m-ASP pathway activity. Based on this analysis, we searched on web databases for mutations that will globally activate silent GENE2s and identified that many GENE2s were reactivated in *fip37* and *mta* mutant lines, therefore suggesting that m6A could be involved in NDT regulation.

We agree with the reviewer's comment that the previous Fig.3A comparing the list of up-regulated loci from two different studies was confusing and not pertinent in the context of the paper. As previously mentioned, we have now executed a more stringent bioinformatic screen on *nerd* RNA-seq data, selecting for changes in strand-specific read density in the 500-bp windows downstream of all annotated genes upon NERD depletion and removing all candidate genes whose expression showed a significant change in *nerd* mutant background. This analysis revealed a more stringent population of NDT genes (n=107) that show a statistically significant downstream readthrough phenotype, but no changes in upstream gene expression (see Fig. 1A). Performing the same type of experiment on *fip37L* RNA-seq data, we also identified a set of 174 loci showing significant readthrough transcription in this mutant, and detected a significantly overlap with these two set of data (see Fig. 3C and supplemental Table S1 and S5), reinforcing the data that FIP37 controls readthrough transcription at NDT loci.

Connection between NERD and m6A is reinforced by our data indicating that readthrough transcription is also observed in the m6A methyltransferase *mta* mutant, by our epistatic

interaction data between NERD and FIP37, and finally by our LC-MSMS and RIP-qPCR data indicating that *nerd* decreases m6A level at a transcriptome and gene specific levels. The list of high confidence m6A peaks was provided in *fip37* (Shen et al. 2012), *mta* (Anderson et al. 2018) and Luo et al. 2014 papers.

5. 3' RACE is not quantitative, not supporting that only specific pA sites are m6A dependent. Furthermore, the same sites seem to be m6A modified in wt and mutant backgrounds.

Response: We agree with the reviewer's comment that cloning and sequencing of 3' RACE products is more qualitative than quantitative. However, we observed that the 3' RACE profiles seen in gel are highly reproducible, showing both quantitative and qualitative changes that reinforce the idea that a decrease polyA site accumulation at GENE1 NDT is associated with mRNA chimera formation. We have clarified the figure, removing the number of clones, just focusing on the position of major polyA sites.

6. m6A reader CPSF30 mutants are not analyzed with global RNA-seq and the conclusion that CPSF30L and FIP37 depletions affect the polyA sites pattern in a quite similar manner is therefore not sufficiently substantiated. In addition, in Fig5C the number of *fip-37* readthroughs is given as 961, while in Figure 3A as over 3000. Figure 5c also suggests that most read-throughs in *fip-37* are different to those caused by disruption of a core poly(A) factor with a YTH domain, hence not supporting the authors' conclusion that NERD, m6A and CPSF30 work together to regulate termination.

Response: We agree with the reviewer and have introduced RNAseq analysis of CPSF30S, CPSF30L and *fip37* that demonstrated that genes showing readthrough transcription in FIP37 mutant significantly overlap with the genes showing readthrough transcription in the plant missing the CPSF30-YTH protein (Fig. 5B,C). We also provide box-plot data reinforcing the claim that FIP37 and CPSF30L act in the same pathway.

7. The authors conclude that m6A is involved in transposon control based on one example, suggesting that NERD does not play a notable role in the control of transcription over transposon sequences.

Response: When we first identified the NDT loci, all GENE2 were annotated as bonafide genes, suggesting that m-ASP was targeting only mRNA chimera. It was only later in the course of our analysis that some GENE2 were reannotated as TE, prompting us to focus our analysis on one GENE-TE candidate.

Concerning TE regulation, we focused our analysis on the AT3G47890-AT3TE71565 loci because it has been known for long that LINE element can escape epigenetic control through the formation of GENE-LINE chimeric transcript, a process also named as transduction. The point we wanted to make here was to show that the m-ASP pathway could be considered as an additional mechanism by which plants could restrict TE expression and propagation. We are conscious that our demonstration is limited to one case, but one should keep in mind that Arabidopsis is by far the worst model to analyze TE dynamics given its low content in TE.

8. Does NERD bind sites that exhibit readthrough?

Response: To date, the exact mechanism by which NERD regulates m6A remains unclear. We tried to chip NERD on NDT loci without success so far. We also failed to detect NERD-FIP37 interaction through co-IP. As indicated in the discussion, recent proteomics data suggest that NERD is associated with the elongating form of Pol II and could facilitate m⁶A

methylation within the Pol II elongation complex.

9. The model is not sufficiently supported by the data provided.

Response: We agree with the reviewer's comment. According to the published m⁶A data and our RACE data, it is clear that cleavage and polyadenylation always occur downstream of m⁶A peak in all NDT loci tested. We have corrected figure 6D in accordance. Concerning the last comment on Fig.3A, we agree that the previous analysis was confusing. In the revised version, we have now provided epistatic, LC-MSMS, and RNA-seq data that reinforce the functional link existing between NERD and FIP37.

10. How do the findings relate to the previous work on NERD by this group? The previous work needs to get introduced.

Response: We agree with the reviewer's comment. We have introduced data indicating that the GW motifs of NERD are not involved in readthrough control at NDT loci (Supplemental Fig. S6), and extensively discussed about the potential function of NERD in m⁶A regulation in the discussion section.

Reviewer #3:

The reviewer's questions the significance of the findings given the known role of m⁶A in plant mRNA stabilization.

Response: We disagree with the reviewer's comment that the number of targets of a given pathway necessarily reflects its importance. Important papers have previously shown that m⁶A controls sex determination in insect by controlling female-specific splicing of the master sex determination factor sex-lethal (Sxl) gene. Meanwhile in human cells, heat stress has been shown to control m⁶A methylation in the 5'UTR of a subset of stress-inducible genes. In regard to our data, we think that the size of m-ASP target set in a particular plant is likely to reflect its levels of genome dynamic and transposon content. However, relaxation of m-ASP-dependent control of a single transposon or the production of nonfunctional bicistronic gene products could actually have deleterious impact of plant fitness.

Finally, I would like to stress the point that our data linking m-ASP to a specific YTH reader, CPSF30-YTH, reinforce the idea that we have identified a m⁶A sub-pathway specific to plants. Considering that plants contain 13 YTH proteins, it is not a surprise to me that only a limited subset of loci will be targeted by each one of these proteins knowing that only a limited set of transcripts is methylated in Wild type plants.

Specific points:

1. Is there a biological significance for the observed chimeric loci?

Response: we do not think that the m-ASP pathway targets necessarily need to have any biological function. On the contrary, we believe that the m-ASP pathway acts as a safeguard mechanism restricting the inappropriate formation, and potential expression, of chimeric mRNA at rearranged NDT loci. This is supported by our new data showing that the NDT AT4G30570 GENE2 is likely to produce a nonfunctional form of GDP-mannose pyrophosphorylase upon mRNA chimera expression (Fig. S5B, C). Moreover, our data indicating that GENE-TE loci are also targeted by m-ASP reinforce the idea that the m-ASP

pathway acts as a safeguard mechanism knowing that GENE-TE chimeric RNA have already been implicated in transposon mobilization in plants.

2. The title needs to get toned-down.

Response: we disagree that the title is over the top since we provide clear evidence that the m-ASP pathway controls the integrity of transcriptome by restricting chimeric mRNA formation at numerous NDT loci.

Dear Dr. Lagrange,

Thank you for submitting your revised manuscript entitled "m6A safeguards transcriptome integrity by restricting RNA chimera formation in Arabidopsis" to Life Science Alliance. The manuscript was assessed by expert reviewers at another journal before, and you provided those reports to us and you revised your manuscript in response to them.

I appreciate your work and the response provided to the concerns previously raised. I could not obtain the reviewer identities of the previous round of review, so opted to seek arbitrary input on your revision from an expert in the field. As you can see below, this expert thinks that while most responses are satisfactory, further text changes are needed to better reflect the data put forward as well as a more quantitative assay to measure read-through transcription (previous concern of reviewer #2 and point 1 of arbitrating reviewer). I would thus like to invite you to submit such a further revised version for publication here. When resubmitting your work, please also pay attention to the following:

- please upload individual figure files without legends, also for the supplementary figures (move legends into manuscript docx text, please)
- please enter all co-authors into our submission system
- please link your profile in our submission system to your ORCID iD, you should have received an email with instructions on how to do so

The typical timeframe for revisions is three months.

Thank you for this interesting contribution to Life Science Alliance. We are looking forward to receiving your revised manuscript.

- A letter addressing the reviewers' comments point by point.
- An editable version of the final text (.DOC or .DOCX) is needed for copyediting (no PDFs).
- High-resolution figure, supplementary figure and video files uploaded as individual files:

See our detailed guidelines for preparing your production-ready images, <http://www.life-science-alliance.org/authors>

B. MANUSCRIPT ORGANIZATION AND FORMATTING:

Reviewer #1 (Comments to the Authors (Required)):

The previous publication that this group put forth on NERD is a highly controversial paper in the field for several reasons, including that the gene was said to be involved in DNA methylation, yet follow-up investigation by the rest of the field has demonstrated that NERD has virtually no effect on DNA methylation. This point has been touched upon by Reviewer #2 and added to the main text, but the authors must be more clear about what the role of NERD is (the title of this protein "Needed for RDR2-Independent DNA methylation"). At that time, their research showed that NERD was part of a very 'hot' topic pathway. Do all of the read-through loci identified in this manuscript have altered DNA methylation in nerd mutants? If not, the authors need to be explicit and clear that their previously identified role for NERD is incorrect, before they propose several years later that NERD is now involved in today's 'hot' topic pathway. For the transposable element that they tested in the last Results section, there was no change on DNA methylation. These inconsistencies are the reason this manuscript has generated so much skepticism from reviewers. If fully performed the points I've listed above and below should clear this up and the paper could be published in LSA.

1. My major criticism of the manuscript is that the authors investigate read-through transcription, but in the main figures never directly detect a longer transcript in a quantitative manner. Thus, the critical molecular experiments are simply lacking the essential pieces of data. As Reviewer #2 commented on, I agree that by no means is 3' RACE a quantitative method. In addition, the only two Northern blots I could find in the manuscript were in the Supplemental Figure S2C, for only two of the genes (and lacked a proper control panel). Sashimi plots of RNA-seq data are notorious for combining neighboring genes and being

inaccurate, just as the process of matching paired ends upon sequencing is highly imperfect. Therefore, I request that when the authors would like to argue that there is read-through transcription (Figure 1,3,4,5,6) they use a quantitative method to directly detect the longer transcript, whether it be by Northern or RNase Protection with an appropriately designed probe.

2. I agree with Reviewer #2 that there is an over-interpretation of the Results. In the last section of the Results, one example of readthrough involving a transposable element (TE) is shown. However, from this one example the authors broadly extrapolate to general TE regulation. There are more TE annotations in the Arabidopsis genome than gene annotations, and we would never find a result on one gene and then claim it to be true for all or most genes. I insist that the authors dial back their claims in the title, abstract, Results heading subtitle, Results section and conclusion to make it clear to the reader that this is a one-TE example, and is unproven at any other TE locations.

3. A second topic in regard to the over-interpretation of the data is the very broadly written title and abstract. Reviewer #2 also focused on this topic, and the authors did not change. Many of the loci identified and not confirmed or from very highly rearranged and possibly mis-annotated regions of the genome. For example, in the topic heading "m6A-assisted polyadenylation controls mRNA chimera...", the only connection that has been made is a correlation, not causation. Even at the end of that section, the authors write "...these results support a model...". The word "controls" should not be used in the topic heading / abstract. I urge the authors to wrangle in their interpretation towards a more narrow and focused conclusion.

4. Many of the points made in the results section focus on overlap between two datasets. When explaining these overlaps, please directly write in p values and FDR into the Results section text with each claim of enrichment / significance.

The previous publication that this group put forth on NERD is a highly controversial paper in the field for several reasons, including that the gene was said to be involved in DNA methylation, yet follow-up investigation by the rest of the field has demonstrated that NERD has virtually no effect on DNA methylation. This point has been touched upon by Reviewer #2 and added to the main text, but the authors must be more clear about what the role of NERD is (the title of this protein "Needed for RDR2-Independent DNA methylation"). At that time, their research showed that NERD was part of a very 'hot' topic pathway. Do all of the read-through loci identified in this manuscript have altered DNA methylation in *nerd* mutants? If not, the authors need to be explicit and clear that their previously identified role for NERD is incorrect, before they propose several years later that NERD is now involved in today's 'hot' topic pathway. For the transposable element that they tested in the last Results section, there was no change on DNA methylation. These inconsistencies are the reason this manuscript has generated so much skepticism from reviewers. If fully performed the points I've listed above and below should clear this up and the paper could be published in LSA.

Response:

As pointed out above, our manuscript was reviewed at another journal by three referees, two of them acknowledging that the finding was potentially interesting, although they also noticed that the mechanistic insights were too limited to meet the standard for publication at this journal. However, they suggested interesting experiments that could improve the mechanistic output of the paper. Based on these suggestions, we made changes and introduced additional data to address most of the reviewers' questions and ask to operate the potential transfer of our study to Life Science Alliance (LSA) to avoid unnecessary delays to publication. We unfortunately notice that the LSA arbitrator is quite harsh, objecting present, but also previous aspects of our team's work.

Regarding the NERD protein, it is correct that its function in DNA methylation in plants remains unclear and that the hypervariable DNA methylation status of the so-called NERD targets makes it possible that the changes we had seen previously might not be due to genotype per se, but to inherited changes in different lab strains of *Arabidopsis*. However, given our previous data indicating that most of NERD targets are newly acquired genomic regions, we have always been convinced that this hypervariability is in itself interesting and that there may be metastable states of methylation explaining why these young loci might oscillate between states of PTGS or TGS. It was in the course of studies that were initially designed to address these questions that we identified the unexpected contribution of the NERD protein in the control of readthrough transcription at psORF and other loci. However, our current data indicate that NERD activity in readthrough control is independent of its previously identified AGO-hook platform and that multiple functions are likely to be associated to this multidomain chromatin-associated protein. We thus do not expect nor want to suggest that the activity of NERD in m⁶A biology and m-ASP pathway has to be mutually exclusive with a putative AGO-hook-related RNA silencing function. Actually, there are many examples in the literature of chromatin-related proteins having a large repertoire of diverse activities (for example the Sin3 master regulator of transcription). We are currently actively pursuing these issues hoping to be able to decipher the roles of NERD in plant biology.

1. My major criticism of the manuscript is that the authors investigate read-through transcription, but in the main figures never directly detect a longer transcript in a quantitative manner. Thus, the critical molecular experiments are simply lacking the essential pieces of data. As Reviewer #2 commented on, I agree that by no means is 3' RACE a quantitative method. In addition, the only two Northern blots I could find in the manuscript were in the Supplemental Figure S2C, for only two of the genes (and lacked a proper control panel). Sashimi plots of RNA-seq data are notorious for combining neighboring genes and being inaccurate, just as the process of matching paired ends upon sequencing is highly imperfect. Therefore, I request that when the authors would like to argue that there is read-through transcription (Figure 1,3,4,5,6) they use a quantitative method to directly detect the longer transcript, whether it be by Northern or RNase Protection with an appropriately designed probe.

Response:

We disagree with the Arbitrator's comment about the fact that we never detected the existence of a longer transcript in a quantitative manner. Upon the initial identification of ITNs loci, we actually addressed the question of the existence of longer transcripts by performing northern blot experiments on two ITNs loci (see Supplementary Figure S2C). These northern blot experiments confirmed the existence of longer transcripts accumulating at a higher level in the nerd-1 mutant background. However, we also noticed in the course of these analyses that the ITN GENE1 and GENE2 loci were duplicated by nature and that it was often very tricky to define probes for northern that would be specific for ITN GENE1, possibly confounding the subsequent interpretation of northern experiments. This is why we decided to perform RT-PCR with primers specific for each ITN GENE1 and GENE2 loci to validate the expression of contiguous readthrough RNAs in all mutant background analyzed in the paper. I would like also to point out that similar RT-PCR approaches have been extensively used by other teams to validate the accumulation of readthrough transcripts in plants, including the fpa mutant that we are analyzing in our paper (see Duc et al. 2013 Plos Genetics). In this regard, we believe that our side-by-side analysis of FPA and NERD chimeric loci somehow validates the specificity of our RT-PCR approach since we could reproduce the results presented elsewhere.

In addition, we believe that our observations are further supported by the Sashimi plots initially requested by previous reviewer 2 that clearly support the accumulation in nerd and fip37 mutants of intergenic cryptic splicing events that cross the GENE1-GENE2 junction. One has also to notice that the level of accumulation of ITN GENE1 transcripts is quite low and that obtaining enough material to perform northern blots (at least 2µg of polyA+ mRNA) from 9d fip37L seedlings is not an easy experiment. Finally, the added value of doing RNase protection assays on ITN remains quite unclear to us given the fact that all ITN chimeric loci present intergenic cryptic splicing events that would lead to cleavage of the RNase protection probe in the intergenic region, therefore precluding the detection of long chimeric RNA forms.

2. I agree with Reviewer #2 that there is an over-interpretation of the Results. In the last section of the Results, one example of readthrough involving a transposable element (TE) is shown. However, from this one example the authors broadly extrapolate to general TE regulation. There are more TE annotations in the Arabidopsis genome than gene annotations, and we would never find a result on one gene and then claim it to be true for all or most genes. I insist that the authors dial back their claims in the title, abstract, Results heading subtitle, Results section and conclusion to make it clear to the reader that this is a one-TE

example, and is unproven at any other TE locations.

Response:

I apologize for my writing that was not clear enough, giving the impression that I was extrapolating the role of m-ASP pathway to general TE regulation. Indeed, the point we wanted to make in Figure 6 was that m6A-assisted polyadenylation could in principle participate to TE repression via the control of the formation of GENE-TE RNA chimera that could occasionally form upon TE insertion in the vicinity of specific gene. We have rewritten and toned down the text.

3. A second topic in regard to the over-interpretation of the data is the very broadly written title and abstract. Reviewer #2 also focused on this topic, and the authors did not change. Many of the loci identified and not confirmed or from very highly rearranged and possibly mis-annotated regions of the genome. For example, in the topic heading "m6A-assisted polyadenylation controls mRNA chimera...", the only connection that has been made is a correlation, not causation. Even at the end of that section, the authors write "...these results support a model...". The word "controls" should not be used in the topic heading / abstract. I urge the authors to wrangle in their interpretation towards a more narrow and focused conclusion.

Response:

We have rewritten and toned down the text according to the arbitrator's comments. Regarding the arbitrator's comments on loci validation and mis-annotation that come from Reviewer #2, we would like to point out that the 7 ITN loci that we validated in the paper have been randomly selected among the short list of high-confidence ITN loci identified in our bioinformatic screen and that experimentally validating all the remaining loci will go far beyond what can reasonably be asked for revision. Then, to answer the mis-annotation point, we have included a phylogenetic study of ITN loci (excluding tandem genes which are repeats of the same gene), showing that GENE1 and GENE2 represent independent gene units and are not alternative 3' end version of GENE1 (Fig. 1B).

Regarding the arbitrator comment on the fact that our work only shows correlation, not causation of the role of m6A in polyadenylation in Arabidopsis, we would like to point out that our conclusions are supported by a set of independent observations, comprising 1) the identification of the roles of both m6A writer and reader components in m-ASP pathway, 2) the implication of m6A binding pocket of CPSF30L, and 3) by our evidences that ITN GENE1 mRNA are m6A methylated and that a reverse correlation exists between levels of m6A modification and levels of readthrough transcription at ITN loci. The only experiment that would certainly demonstrate the causative role for m6A in m-ASP pathway would be to mutate m6A sites at ITNs. However, this type of experiment has hardly been performed in animal and yet never done in plants.

4. Many of the points made in the results section focus on overlap between two datasets. When explaining these overlaps, please directly write in p values and FDR into the Results section text with each claim of enrichment/significance.

Response:

Statistical analyses have been performed as suggested by the arbitrator and their results

have been introduced in the Result section. Hypergeometric statistical tests were performed using the dhyper function in R (3.4.4). These analyses indicate that the overlaps of genes experiencing readthrough transcription in nerd-1 and fip37 or cpsf30-1, cpsf30-3 and fip37L mutant backgrounds are highly significant.

Thank you for submitting your revised manuscript entitled "The m6A pathway protects the transcriptome integrity by restricting RNA chimera formation in plants". I appreciate your response to the arbitrator's comments and the introduced changes, and I would be happy to publish your paper in Life Science Alliance.

Please log into our system one more time to fill in the electronic license to publish form. Your new manuscript number will be LSA-2019-00393-TRR, please make sure to move all manuscript files to this new stage (single click process).

A. FINAL FILES:

B. MANUSCRIPT ORGANIZATION AND FORMATTING:

We encourage our authors to provide original source data, particularly uncropped/-processed

electrophoretic blots and spreadsheets for the main figures of the manuscript. If you would like to add source data, we would welcome one PDF/Excel-file per figure for this information. These files will be linked online as supplementary "Source Data" files.

Dear Dr. Lagrange,

Thank you for submitting your Research Article entitled "The m6A pathway protects the transcriptome integrity by restricting RNA chimera formation in plants". It is a pleasure to let you know that your manuscript is now accepted for publication in Life Science Alliance. Congratulations on this interesting work.

DISTRIBUTION OF MATERIALS:

Again, congratulations on a very nice paper. I hope you found the review process to be constructive and are pleased with how the manuscript was handled editorially. We look forward to future exciting submissions from your lab.

Sincerely,

Andrea Leibfried, PhD
Executive Editor
Life Science Alliance
Meyerohofstr. 1
69117 Heidelberg, Germany
t +49 6221 8891 502
e a.leibfried@life-science-alliance.org
www.life-science-alliance.org